

# Semi-supervised associative classification using ant colony optimization algorithm

Hamid Hussain Awan and Waseem Shahzad

Department of Computer Science, National Unibersity of Computer and Emerging Sciences Islamabad, Islamabad, Pakistan

## ABSTRACT

Labeled data is the main ingredient for classification tasks. Labeled data is not always available and free. Semi-supervised learning solves the problem of labeling the unlabeled instances through heuristics. Self-training is one of the most widely-used comprehensible approaches for labeling data. Traditional self-training approaches tend to show low classification accuracy when the majority of the data is unlabeled. A novel approach named Self-Training using Associative Classification using Ant Colony Optimization (ST-AC-ACO) has been proposed in this article to label and classify the unlabeled data instances to improve self-training classification accuracy by exploiting the association among attribute values (terms) and between a set of terms and class labels of the labeled instances. Ant Colony Optimization (ACO) has been employed to construct associative classification rules based on labeled and pseudo-labeled instances. Experiments demonstrate the superiority of the proposed associative self-training approach to its competing traditional self-training approaches.

# INTRODUCTION

Semi-supervised learning has become an attractive area of research in various application domains of data mining where fully labeled data is not available. It has got the attention of researchers in recent years in domains like bio-informatics and web mining where only a small portion of data is labeled (*Zhu & Goldberg, 2009a*). SSL is an extension of supervised learning and unsupervised learning.

Supervised learning is a mapping of data instances to their appropriate class labels. Classification is a supervised learning task that maps or attempts to map the instances to their respective classes. During training, classifiers learn the knowledge of predicting the correct labels of given instances. The performance of the classifiers is tested on an unseen set of instanced called the test set to measure its performance. Popular classifiers include Decision Tree (*Quinlan, 1993*), Naive Bayesian classifier (*Rish, 2001*), Ant Miner (*Parpinelli, Lopes & Freitas, 2002*), etc. Decision tree and Ant Miner are examples of rule-based classifiers. Rule-based classifiers construct classification rules that are human-interpretable in *if <antecedent> then <consequent>* form. A classification rule consists of two parts, *antecedent* and *consequent*. The antecedent is a collection of attribute values

Corresponding author
Hamid Hussain Awan,
hamidawan@gmail.com

(terms) that when occur in combination, belong to exclusively one class label. The consequent is the class label of the antecedent. A *Dataset* is split into *training set* and *test set*. The training set is used to train the classifier to learn the mapping between an instance and its class label in the training set. Decision trees use *information gain* as the basis for splitting the data into branches, where each leaf node is a class. The path from the root node to each leaf forms a *classification rule*. Once, the tree is constructed, the test set is used to *generalize* the decision tree by evaluating the classification accuracy. Classification is used for solving various real-life problems like disease detection based on symptoms, fault detection in communication systems, classification of crops, detection of fake news, etc. (*Akhter et al., 2021*).

Unsupervised learning discovers groups called clusters in the data based on similarity among data instances. K-mean clustering is the most popular clustering technique (*Tatsumi, Sugimoto & Kusunoki, 2019*). The goal of the unsupervised learning is to keep the similar instances in the same cluster and different instances in different clusters. Various distance measures like *Euclidean distance* and *Manhattan distance* are used to determine similarity among instances of the given dataset. Clustering aims at minimizing intra-cluster distance and maximizing inter-cluster distance, so that there is a clear distinction between items of different clusters and maximum similarity of items (data instances) in the same cluster.

Association Rule Mining (ARM) is another unsupervised approach that discovers frequent patterns (itemsets) in the given data (*Gazi, 2010*). Market Basket Analysis is one of the most famous problems in ARM. Popular ARM algorithms include *apriori* algorithm and *Frequent Pattern Tree* mining algorithm (*Nguyen et al., 2018*; *Narvekar & Syed, 2015*). Two measures known as *support* and *confidence* are the main ingredients of the ARM. The support is based on the frequency of an itemset in the dataset while confidence is the measure of association among itemsets. Association of itemsets is the relative frequency of an itemset to another itemset. More frequently two or more itemsets occur with each other in the dataset, more associativity exists in the given dataset. An association rule is like a classification rule but since there is no class label involved in ARM, both the antecedent and the consequent are itemsets (attribute-values) in the dataset.

Associative Classification combines frequent-pattern discovery of Association Rule Mining (ARM) (*Nguyen et al., 2018*; *Agrawal & Srikant, 1994*) with classification. The objective of ARM is to discover mutual association of items in itemsets for prediction of inter-dependence of items in given transactions (instances). Frequent patterns are discovered to analyze whether a specific pattern of items is dependent on existence of another pattern (*Narvekar & Syed, 2015*). The ARM works due to its *associativity property*. This property is the measure of association among itemsets (patterns) in the given data. The frequency of patterns is also called the *support* and the associativity of a pattern is called the *confidence*. The difference between associative classification and ARM is that in an *associative classification rule* consequent is always a class label (*Aburub & Hadi, 2018*). Associative classification has shown better performance than non-associative

classifiers (*Shahzad & Baig, 2011*). A detailed description of terms related to associative classification is presented in Basic terms of Associative Classification.

Semi-supervised Learning (SSL) is an emerging technique that involves learning from a smaller amount of labeled data and then using the learned knowledge to label the unlabeled data (*Zhu, Yu & Jing, 2013*). There are two types of SSL. One is called *Semi-Supervised Classification* in which SSL is used for classification purpose. The other type is called *Semi-Supervised Clustering* or *constrained clustering* which is used to improve clustering performance with the help of labeled instances (*Li et al., 2019*; *Triguero, Garca & Herrera, 2015*). Semi-Supervised classification (SSC) is the subject of this article. Semi-supervised classification training consists of two steps, *training* and *pseudo-labeling*. A detailed description of SSL mechanism and definition of terms is explained in Basic terms in SSL.

The aim of the supervised learning is to map data instances to target patterns (class labels) (*Fu et al., 2020*), while unsupervised learning aims at grouping data instances on the basis of mutual similarity (*Tatsumi, Sugimoto & Kusunoki, 2019*). Whereas semi-supervised learning is a hybridization of both the supervised learning and the supervised learning. Training of a typical semi-supervised classification model consists of two iterative steps. In the first step, the model is trained on labeled data (supervised learning), while in the second step pseudo-labeling is performed to assign label to some of the unlabeled instances based on similarity with labeled instances or based on classification rules (unsupervised learning). This two-step proses is repeated until all unlabeled instances are pseudo-labeled (*Triguero, Garca & Herrera, 2015*). Supervised learning (classification) is performed on data where all instances are labeled. Unsupervised learning (clustering and ARM) does not need class labels for its operation. Semi-supervised learning is used on data containing both the labeled and unlabeled instance.

Self-labeling is one of the most widely-used approach to perform SSC (*Yarowsky, 1995a*; *Li & Zhou, 2005*). It consists of two phases. In the first phase, labeled data is used to train traditional classifiers (*e.g.*, C4.5 *Quinlan, 1993*) to find a mapping between data distribution and class labels. This knowledge is then used in the second phase to assign labels to unlabeled instances of the data set. There are two slightly different ways of training and assigning labels in semi-supervised learning. One is called the *inductive learning* in which only labeled instances are used during training and unlabeled instances are assigned labels only, without being part of the training. The other approach is called the *transductive learning* in which iterative procedure is followed to label the selected unlabeled instances and then use them as part of the labeled set to label remaining unlabeled instances (*Zhu, Yu & Jing, 2013*). There are two types of self-labeling in literature named *self-training* and *co-training* (*Ling, Du & Zhou, 2009*).

Self-training employs one classification algorithm to construct classification rules using labeled instances. It is retrained on *extended* labeled set of instances (see Definition 3) containing both the labeled and pseudo-labeled instances to refine classification model. Self-training doesn't make any specific assumptions about the underlying dataset except that it assumes its classification model is correct (*Zhu, Yu & Jing, 2013*).

Co-training (*Fujino, Ueda & Saito, 2008*) splits the underlying datasets vertically. Each partition is called a view. Each view is used to train a traditional classifier independent of other views (*Blum & Mitchell, 1998*). After training of classifier on all views, the classifiers share their model with each other to teach each other about the most confident predictions. Co-training assumes that the underlying dataset can be split into multiple conditionally independent views (*Jiang, Zhang & Zeng, 2013*).

Ant Colony Optimization (ACO) is a *meta heuristic* inspired by social behavior of ants. It is a stochastic search approach based on ants' foraging behavior. Real ants communicate with each other with the help of a chemical called *pheromone*. Each ant deposits pheromone while moving in search of food (*Mohan & Baskaran, 2012*; *Chen et al., 2020*). Unlike mathematical models which follow greedy search approach, ACO performs probabilistic random search which helps the model avoid from converging into local optimum solution. Instead, ACO provides a diverse set of solutions which may not look good initially but they evolve and an optimal or a near-optimal solution is discovered (*Parpinelli, Lopes & Freitas, 2002*). ACO does not guarantee optimum solution, but it *attempts* to discover *optimum* or *near-optimum* solution to the given problem. Despite of not providing the guaranteed optimal solution, ACO has been successfully applied in various optimization problems such as Constraint Satisfaction Problem (*Guan, Zhao & Li, 2021*) and data mining problems to show promising results outperforming deterministic greedy algorithms (*Shahzad & Baig, 2011*). A comprehensive description of how ACO works is explained in are Ant Colony Optimization (ACO).

The motivation for the proposed research is the combination of impressive performance of associative classification and diversity of the Ant Colony Optimization (ACO) algorithm. Associative classification makes use of association among frequent pattern before predicting the class labels of instances. Such patterns may exist independent of the data being labeled or unlabeled. If associative classification rules have been discovered in the labeled data, such patterns may also exist in the unlabeled data. Same classes may be assigned to similar patterns in the unlabeled data. Thus the task of assigning labels to unlabeled instances would be simpler and more robust than comparing each unlabeled instance to labeled instances every time for assigning the label. The use of ACO for discovering frequent patterns in classification and associative classification in labeled data has shown promising results (*Parpinelli, Lopes & Freitas, 2002*; *Shahzad & Baig, 2011*). Therefore a combination of associative classification and ACO is expected to construct a more accurate and robust semi-supervised classifier.

This article proposes a transductive *self-training* Semi-Supervised Classification by exploiting mutual association among attributes-values of underlying data. The proposed approach employs associative classification using ACO for self-training. This technique is named Self-Training using Associative Classification using Ant Colony Optimization (ST-AC-ACO). The reason for choosing self-training is that it doesn't make any assumption about the data distribution. It only makes the assumption that its class predictions or pseudo-labeling are correct (*Witten, Frank & Hall, 2011*; *Blum & Mitchell, 1998*). Unlike

traditional semi-supervised self-training algorithms, ST-AC-ACO employs associative classification which adds a level of confidence for more accurate label prediction (*Hadi, Al-Radaideh & Alhawari, 2018*; *Venturini, Baralis & Garza, 2018*). Associative classification as self-training technique is new to our knowledge and experiments show that it has outperformed existing self-training algorithms (see Experimental Results). The performance of the proposed technique is compared with five state-of-the-art top performing techniques. The significance of results of classification accuracy is tested using non-parametric Wilcoxon Signed Rank Test (*Garca et al., 2010*) for four different ratios of labeled data to verify the results. The *Kappa* statistics are used to evaluate the performance of ST-AC-ACO in comparison to its competing algorithms. The following contributions have been made by the proposed approach:

- A novel transductive self-training technique by utilizing associative classification rules.
- Derivation of equations for calculation of support and confidence of associative classification rules in pseudo-labeled instances.

The rest of the article is as follows: Background presents the preliminary background of SSL and ACO, Related Work presents related work, Proposed Methodology explains our proposed technique, Experimental Results demonstrates experimental results and comparison of the proposed technique with other self-training techniques and Conclusion concludes the article.

## BACKGROUND

This section presents the basic definitions of terms related to SSL, Associative Classification and ACO.

### Basic terms in SSL

**Definition 1** Labeled set $L$ is a subset of dataset $D$ consisting of the data instances which have class labels.

**Definition 2** Unlabeled set $U$ is a subset of $D$ consisting of the data instances which don't have class labels.

Mathematically:

$$D = L \cup U \tag{1}$$

Moreover

$$L \cap U = \phi \tag{2}$$

**Definition 3** Extended labeled set $EL$ is a sub set of $D$ which is initially $L$ (*i.e.*, $EL = L$). Instances from U are assigned labels and included in the $EL$. Such instances that are assigned labels by some heuristic are called pseudo-labeled instances. When all the instances from $U$ are labeled and added to $EL$, the $EL$ becomes equal to $D$ (*Triguero et al., 2014*; *Zhu, Yu & Jing, 2013*; *Triguero, Garca & Herrera, 2015*).

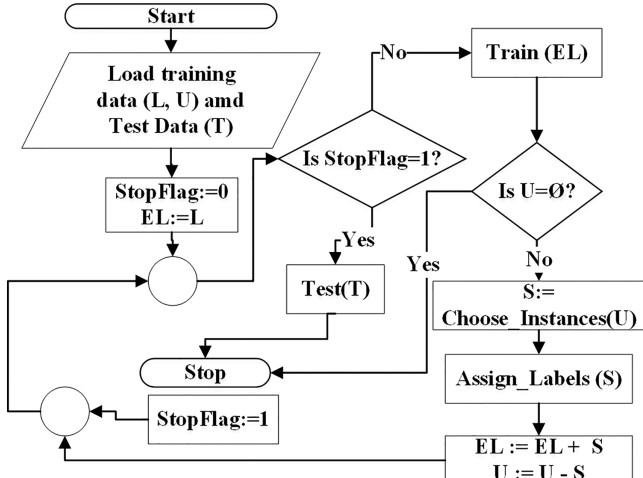

**Figure 1 Flowchart of semi-supervised classification.**

**Definition 4** Enlargement of *EL* is the process of selecting instances from *U*, assigning them labels and moving them from *U* to *EL*. There are three proposed mechanisms for EL enlargement (*Triguero, Garca & Herrera, 2015*). They are:

- **Incremental:** A fixed number of instances are chosen from *U* to move to *EL* after assigning most appropriate class to each instance (*Jiang, Zhang & Zeng, 2013*).
- **Batch:** Each instance is evaluated under additional criteria before being added to *EL*. The basic criterion is the measure of confidence or similarity of an instance to some labeled instances for assigning the most appropriate class. After each instance is labeled, all pseudo-labeled instances are moved to *EL* in a single batch.
- **Amend:** Pseudo-labeled instances are continuously monitored and re-evaluated to measure any mis-labeling. Mis-labeled pseudo-labeled instances are re-labeled. This technique is more accurate than others but its much higher time complexity makes it impractical for application (*Li & Zhou, 2005*).

The goal of SSC is to first learn from the labeled data, apply the learned knowledge to extend the labeled data by pseudo-labeling and then testing the results on test data. According to the flowchart in Fig. 1, the self-training algorithm reads the training data and the test data (*T*). The training data consists of labeled data (*L*) and unlabeled data (*U*). Extended labeled data (*EL*) is initialized with the data in *L*. The classifier is trained on *EL* and a specific (randomly selected) instances from *U* are picked for pseudo-labeling. Each instance is assigned a class label based on classifier rules that were constructed during training on *EL*. After pseudo-labeling, the picked instances are moved into *EL*. If the training mode is *inductive*, all unlabeled instances can be pseudo-labeled in one iteration because pseudo-labeled instances are not used in training. But in *transductive* mode, unlabeled instances are iteratively pseudo-labeled and moved from *U* to *EL* and are used in training in subsequent iterations. This is depicted in the flowchart of Fig. 1.

The process of pseudo-labeling terminates when all unlabeled instances are pseudo-labeled and added to *EL* set. Finally testing is performed on the *test set* (*T*).

## Basic terms of Associative Classification

**Definition 5** Pattern is an associative classification rule that states association of an itemset *X* with a class label *Y*. The antecedent of a pattern is *X* while the consequent is *Y* (*Agrawal & Srikant, 1994*; *Hadi, Al-Radaideh & Alhawari, 2018*).

**Definition 6** Support of a pattern (*X* => *Y*) is calculated as:

$$Supp(X => Y) = P(X \cup Y) \tag{3}$$

where *Supp*(*X* =>*Y*) denotes the support of pattern *<if X then Y >* while $P(X \cup Y)$ represents the probability of occurrence of itemset *X* with class label *Y* (*Hadi, Al-Radaideh & Alhawari, 2018*; *Nguyen et al., 2018*).

**Definition 7** Confidence of a pattern (*X* => *Y*) (*Venturini, Baralis & Garza, 2018*; *Hadi, Al-Radaideh & Alhawari, 2018*) is calculated as:

$$Conf(X => Y) = P(Y|X) \tag{4}$$

where *Conf*(*X* =>*Y*) denotes confidence of the pattern *X* =>*Y* while $P(Y|X)$ represents the probability of occurrence of class label *Y* given the occurrence of itemset *X* (*Agrawal & Srikant, 1994*).

In associative classification, attribute values (terms) and their combinations are called *patterns*. The support for patterns is calculated to discover the frequent patterns among them. The class labels are combined with frequent patterns to construct the associative classification rules in which the antecedent is a pattern and the consequent is a class of each rule. The confidence of each rule is calculated and confident rules are added to the rule list.

To get a better understanding of associative classification, a sample hypothetical dataset has been presented in Table 1. The dataset is about participation of people in a social campaign.

There are three features of each person namely, *Age group*, *Gender* and *Social* (socialization) while the *Participate* denotes the participation of the person in social campaigns (*Yes* means the person participated in social campaigns).

In the first step in associative classification is to discover frequent patterns. A frequent pattern is one whose support meets *minimum support* user-defined threshold. In simple words, the support of a of a pattern is obtained by dividing its count of occurrences in the dataset by the total number of dataset instances. For instance, in the sample dataset, {*Gender = Male, Social = Introvert*} is a pattern which occurs 3 times. The dataset consists of 10 instances, hence the support of the pattern is 0.3. The confidence of each frequent pattern is calculated against each class label (see Definition 7). For this purpose, the support of the frequent pattern in each class is divided by the pattern's support. The pattern {*Gender = Male, Social = Introvert*} occurs 2 times with the class label *Yes* and once with the class label *No*. Thus the associative classification rule {*Gender = Male, Social = Introvert*} => *Participate = Yes* has a support of 0.2 in the dataset while its confidence is

**Table 1 Sample dataset related to participation in a social campaign.**

| Age group | Gender | Social | Participate |
|---|---|---|---|
| Teen | Male | Extrovert | Yes |
| Teen | Female | Introvert | No |
| Mature | Male | Extrovert | Yes |
| Mature | Male | Introvert | No |
| Old | Female | Extrovert | Yes |
| Mature | Male | Introvert | No |
| Teen | Male | Extrovert | Yes |
| Mature | Female | Introvert | Yes |
| Teen | Female | Extrovert | No |
| Mature | Male | Introvert | Yes |

**Table 2 Support and confidence of single-term associative classification rules of the sample dataset.**

1-Term patterns (Min. support = 0.2, Min confidence = 0.6)

| Pattern | Support | Is frequent? | Participate | Confidence | Is confident? |
|---|---|---|---|---|---|
| Age_Group = Mature | 0.5 | Frequent | No | 0.40 | |
| | | | Yes | 0.60 | Confident |
| Age_Group = Old | 0.1 | … | | | |
| Age_Group = Teen | 0.4 | Frequent | No | 0.50 | |
| | | | Yes | 0.50 | |
| Gender = Female | 0.4 | Frequent | No | 0.50 | |
| | | | Yes | 0.50 | |
| Gender = Male | 0.6 | Frequent | No | 0.33 | |
| | | | Yes | 0.67 | Confident |
| Social = Extrovert | 0.5 | Frequent | No | 0.20 | |
| | | | Yes | 0.80 | Confident |
| Social = Introvert | 0.5 | Frequent | No | 0.60 | Confident |
| | | | Yes | 0.40 | |

$\frac{0.2}{0.3} = 0.67$. The confidence of {*Gender = Male, Social = Introvert*} *=> Participate = No* is 0.33. Let us assume the minimum support threshold in the sample dataset is 0.2 and minimum confidence is 0.6. Thus the classification rule {*Gender = Male, Social = Introvert*} *=> Participate = Yes* is confident while {*Gender = Male, Social = Introvert*} *=> Participate = No* is unconfident.

In case of the *Apriori* algorithm (*Agrawal & Srikant, 1994*), the process is repeated for an *exhaustive* combination of all terms against each class. Terms or their combinations are also referred to as *patterns*. A meta-heuristic like ACO tries to avoid exhaustive search for associative rules by exploring the search space in a *guided random* way.

Table 2 lists the 1-term patterns and their supports The rules having support of 0.2 or more are frequent. Confidence of each rule resulted from respective frequent pattern is

**Table 3 Support and confidence of 2-term associative classification rules of the sample dataset.**

2-Term patterns (Min. support = 0.2, Min confidence = 0.6)

| Pattern | Support | Participate | Confidence | Is Confident? |
|---|---|---|---|---|
| Age_Group = Mature, Gender = Female | 0.1 | | | |
| Age_Group = Mature, Gender = Male | 0.4 | No | 0.50 | |
| | | Yes | 0.50 | |
| Age_Group = Old, Gender = Female | 0.1 | | | |
| Age_Group = Teen, Gender = Female | 0.2 | No | 1.00 | Confident |
| Age_Group = Teen, Gender = Male | 0.2 | Yes | 1.00 | Confident |
| Age_Group = Mature, Social = Extrovert | 0.1 | | | |
| Age_Group = Mature, Social = Introvert | 0.4 | No | 0.50 | |
| | | Yes | 0.50 | |
| Age_Group = Teen, Social = Extrovert | 0.3 | No | 0.33 | |
| | | Yes | 0.67 | Confident |
| Age_Group = Teen, Social = Introvert | 0.1 | | | |
| Gender = Female, Social = Extrovert | 0.2 | No | 0.50 | |
| | | Yes | 0.50 | |
| Gender = Female, Social = Introvert | 0.2 | No | 0.50 | |
| | | Yes | 0.50 | |
| Gender = Male, Social = Extrovert | 0.3 | Yes | 1.00 | Confident |
| Gender = Male, Social = Introvert | 0.3 | No | 0.67 | Confident |
| | | Yes | 0.33 | |

calculated with each class. The rules with confidence value of 0.6 or more are considered confident rules. Confident rules are retained while others are discarded. Notice that {*Age_Group = Old*} is an infrequent pattern. So it is neither an associative rule nor it will be used for construction of multi-term patterns. On the other hand, 4 associative classification rules have been discovered. The rule {*Social = Extrovert*} => *Participate = Yes* is the most confident with confidence of 0.8 and support of 0.5. To keep the table width in the page limit, only class labels (consequents) of the associative classification rules have been mentioned. The antecedent is the same as the pattern itself.

Table 3 lists 2-term patterns and their supports. The resultant associative classification rules are also displayed with their confidence. A total of 3 out of 5 associative classification rules have confidence of 1.0 which means that all such patterns belong to only one class in the given dataset. To keep the table inside the page limit, the *Is Frequent* column has been removed from the table. Patterns with support greater than or equal to the minimum support are frequent patterns.

Table 4 shows the list of 3-term patterns and resulting associative classification rules. Due to length of the patterns, attribute names are not shown in patterns. The notable thing here is that support of patterns in this table has been decreased and only two patterns are frequent. But both of them are confident. This shows the power of associativity of frequent pattern mining (*Shahzad & Baig, 2011*). Longer patterns tend to show stronger

**Table 4 Support and confidence of 3-term associative classification rules of the sample dataset.**

3-Term patterns (Min. support = 0.2, Min confidence = 0.6)

| Pattern | Support | Participate | Confidence | Is Confident? |
|---------|---------|-------------|------------|---------------|
| Mature, female, introvert | 0.1 | | | |
| Mature, male, extrovert | 0.1 | | | |
| Mature, male, = introvert | 0.3 | No | 0.67 | Confident |
| | | Yes | 0.33 | |
| Teen, female, extrovert | 0.1 | | | |
| Teen, female, introvert | 0.1 | | | |
| Teen, male, extrovert | 0.2 | Yes | 1.00 | Confident |

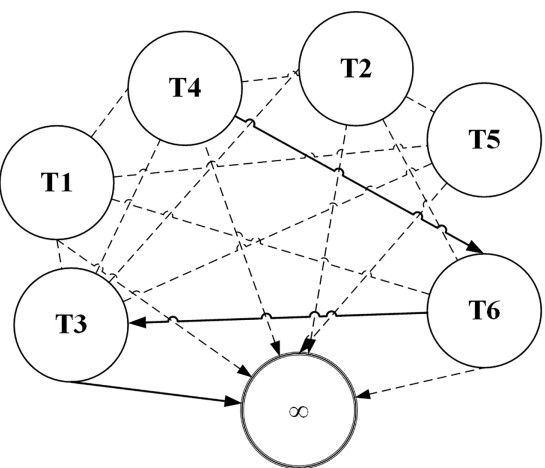

**Figure 2 ACO representation as a graph data structure.**

confidence but have lower support. So if longer frequent patterns are discovered, more confident associative classification rules are constructed.

## Ant colony optimization (ACO)

A problem can be represented as a 2-dimensional graph data structure in ACO algorithm (*Parpinelli, Lopes & Freitas, 2002*). As real ants use pheromone for mutual communication, the artificial ants also have a *pheromone* stored as a global repository to guide other ants about the optimum path(s). The *pheromone* and the *heuristic* are used to calculation of the *selection probability* of a *path* in the graph by an ant. Heuristic is a problem-dependent measure which is usually set for example in shortest-path finding problems as the inverse of the distance between two nodes of a graph. The lower the value of a distance means the higher the heuristic value.

ACO has most of its applications on categorical data sets. Terms (attribute values) are represented by *nodes* and selection probabilities of a term being chosen are represented by edges of the graph as shown in Fig. 2. The higher the selection probability of a term, the higher it is likely to be selected by an ant. Terms of the same attribute can't be

connected in the graph because only one term of an attribute can be selected in a pattern. For example $T1$ and $T2$ belong to same attribute in the given figure. To understand more clearly, consider the sample dataset in Table 1 and suppose $T1$ represents the term *Gender = Male* while $T2$ represents *Gender = Female*. Obviously, a classification rule can't contain both the terms. Otherwise its coverage will be zero as no instance contains these two terms simultaneously.

The node marked with $\infty$ is the *sink* node which can be selected after selection of at least one term. The search process of an ant is terminated when an ant reaches sink node. As shown in the figure, let us assume that an ant selects the term $T4$ with the help of probabilistic random search. After $T4$ has been selected, the selection probabilities of other terms are considered for selection from viewpoint of $T4$ (all nodes leaving $T4$ node in the graph). According to the figure, the ant chose $T6$ and from there it chose $T3$. From $T3$, the ant picked the sink node. A sink node is selected when the random number used for selecting a node matches the selection probability of the sink node. Thus the ant searched a *path T4 − T6 − T3*. Since the nodes are terms of the classification dataset, they map to the antecedent of a classification rule. Generally, the antecedent is evaluated for coverage and is assigned the consequent of the class label with which it has the highest frequency. The solid lines represent the path selected by the ant during its search process. The dashed lines represent the unvisited edges by the ant.

**Definition 8** Pheromone in ACO acts for the material deposited by real ants when searching for food. It is used to guide other ants during search of the most optimum paths. Pheromone values can be initialized to zero or some arbitrary value between 0 and 1. A more appropriate way of initializing the pheromone values is given in Eq. (5) (*Shahzad & Baig, 2011*).

$$\tau_{ij} = \frac{1}{\sum_{i \in A} b_i} \tag{5}$$

where $\tau_{ij}$ denotes the pheromone value between nodes (terms) $i$ and $j$, $A$ represents set of attributes while $b_i$ represents number of terms of the $i$th attribute.

**Definition 9** Heuristic is a problem-dependent value which usually evaluates the fitness of the solution component. An example heuristic can be the weight of the edge between two nodes. Ant Miner algorithm (*Parpinelli, Lopes & Freitas, 2002*) uses *entropy* measure used in information theory. Heuristic value is calculated using Eqs. (6) and (7).

$$P_{ij} = P(w|A_i = V_{ij}) \tag{6}$$

$$H(W|A_i = V_{ij}) = -\sum_{w \in C} P_{ij} log(P_{ij}) \tag{7}$$

where $H$ represents heuristic value between nodes (terms) $i$ and $j$, $w$ represents the class label, $C$ represents set of class labels, $A_i$ represents the $i$-th attribute, $V_{ij}$ represents $j$-th value of $A_i$ and $P(w|A_i = V_{ij})$ represents the conditional probability of class label $w$ given that $A_i = V_{ij}$ has occurred.

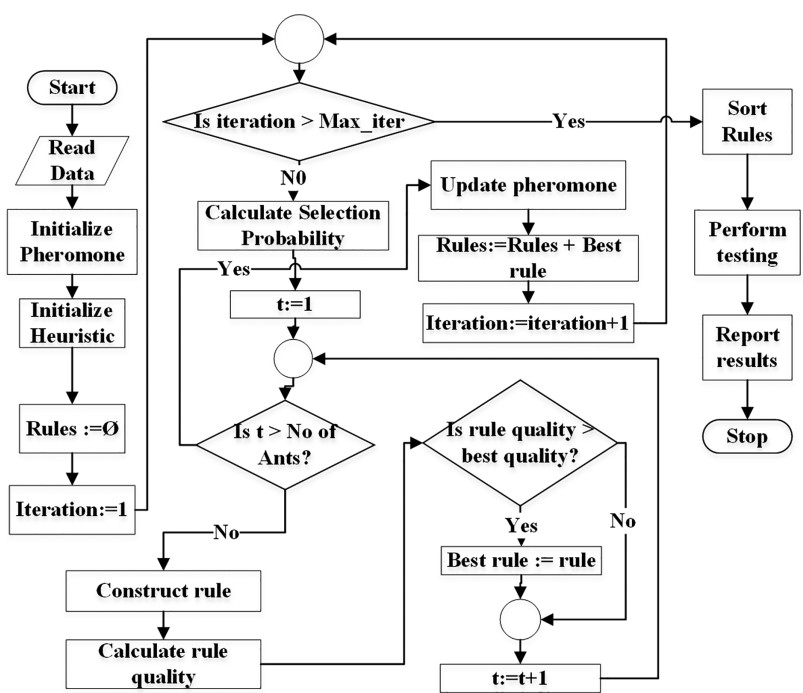

**Figure 3 Flowchart of the generic ACO algorithm for classification.**

**Definition 10** Selection probability is the guideline for ants to search for most optimal paths. Probability is a combination of pheromone and heuristic values (*Guan, Zhao & Li, 2021*; *Mohan & Baskaran, 2012*)

$$P_{ij} = \frac{[\tau_{ij}]^\alpha [\eta_{ij}]^\beta}{\sum_{v \in V} [\tau_{iv}]^\alpha [\eta_{iv}]^\beta} \tag{8}$$

where $P_{ij}$ denotes probability of selecting node $j$ from node $i$ and vice versa, $\tau_{ij}$ represents pheromone between nodes $i$ and $j$, while $\eta_{ij}$ represents problem-dependent heuristic value. Parameters $\alpha$ and $\beta$ represent the *weights* of pheromone and heuristic values respectively.

**Definition 11** Pheromone of search paths evaporates (decreases) over time. Pheromone evaporation rate $\rho$ is usually kept constant in ACO and is a user-defined parameter. Its value is kept around 0.1 (*Parpinelli, Lopes & Freitas, 2002*).
**Definition 12** The increase in the pheromone values of paths with best results is called pheromone update. This update increases the selection probability of edges in best paths for future iterations by ants (*Shahzad & Baig, 2011*).
**Definition 13** ACO algorithm is terminated when either a user-defined maximum number of iterations has been executed or the best searched path hasn't been changed for a (user-defined) number of iterations (*Mohan & Baskaran, 2012*).

Figure 3 demonstrates the flowchart of the generic ACO algorithm for classification (*Parpinelli, Lopes & Freitas, 2002*). The *pheromone* matrix is usually a 2-dimension

square matrix of size equal to the number of terms in the dataset. The *heuristic* and *probability* matrices would have the same size as of *pheromone*. pheromone matrix is initialized using Eq. (5) while heuristic is initialized using Eq. (7). The *Rules* list is initially empty and is used to store the classification rules during training.

The algorithm is executed for *Max_iter* (user-defined parameter) number of times. Selection probability pf all terms is calculated before an ant *t* starts constructing its rule. Each ant constructs a rule using the mechanism demonstrated in Fig. 2.

Each ant constructs its rule on its turn. The total number of ants *No_of_ants* is a user = defined parameter. Its value is usually kept between 10 to 100 (*Parpinelli, Lopes & Freitas, 2002*).

Once the ant *t* finishes its journey (reaches the sink node), the quality of its rule is calculated. There are various measures for calculations of the rule quality (*Parpinelli, Lopes & Freitas, 2002*; *Mohan & Baskaran, 2012*). The classification accuracy of a rule can also be used as quality of the rule. There are two aspects of a rule itself, the *coverage* and the quality. The coverage refers to the number of instances the antecedent of the rule matches to in the dataset. The class label of the majority of the covered instances is set as the consequent of the rule. The quality (*e.g.*, accuracy) refers the ratio of the number of the instances correctly covered by the rule to the total number of instances covered by the rule.

After all ants construct their rules, the pheromone values are *evaporated* depending on the pheromone evaporation rate ($\rho$) set by the user (see Definition 11) according to Eq. (17). The evaporation process moderates the negative impacts of a non-optimal path selection (by an ant) in future iterations.

The rule with the best quality is used to update (increase) pheromone values of the terms used in the rule (see Eq. (18)). This means that only ants with the best rule quality are allowed to update the pheromone values.

Since the probability of term selection depends on pheromone values (see Eq. (8)), updated pheromone modifies the selection probabilities at the start of new iteration.

The rule with the best quality in an iteration is added to the rule list (*Rules*). This process continues until the number of iterations reaches the user-defined limit (*max_iterations*). Finally the rules are pruned duplicate rules are removed from the rule list. This concludes the training. The testing is performed on the test set and results are reported.

## RELATED WORK

*Shahzad & Baig (2011)* proposed a robust classifier using associative classification using Ant Colony Optimization for labeled data sets. This model uses the *select class first* approach to construct rules for a selected class only. Rules for all the classes are constructed by choosing classes one-by-one. This technique experimentally showed much better accuracy than its competitors. This approach has been applicable to supervised classification problems only.

*Aburub & Hadi (2018)* developed an associative classification algorithm for prediction of existence of underground water at a given place. Again this algorithm has been developed for associative classification of fully-labeled data.

Associative classification approaches have been applied for labeled datasets only and there exists no work on associative classification for semi-supervised learning of datasets containing unlabeled instances according to our knowledge.

*Chen et al. (2020)* utilized Ant Colony Optimization for controlling the pollutant information on social media in *Chen et al. (2020)*. The problem was formulated as a *bi-objective* problem. The two objectives specified were the maximization of effect of the control and the minimization of the cost of the control. The results of the proposed approach showed competitive results with respect to control effect maximization when compared to the best techniques for this object, while it outperformed its competitors in minimizing cost of the control.

*Zhu & Goldberg (2009b)* put forward the initial formalization and classification of Semi-Supervised Learning (SSL) techniques.

*Triguero, Garca & Herrera (2015)* presented a taxonomic study of self-leveling techniques in Semi-Supervised Classification. This study provides a critical review of the self-labeling methods and also presents software tools for self-labeling SSC in *Triguero, Garca & Herrera (2015)*. The main contribution of this research work includes proposing of new taxonomy of self-labeling methods, analysis and deduction of transductive and inductive capabilities of the self-labeling methods, and establishing an experimental methodology of the state-of-the-art self-labeling techniques along with the introduction of self-labeling module for KEEL software. The problem with this approach is that it compares self-training and co-training versions of traditional classification algorithms and no additional measure is used in classification process like feature selection or associative classification.

*Li & Zhou (2005)* proposed a self-labeling technique called SETRED that employs *amending* mechanism of extending the labeled set by reviewing the labeling process of pseudo-labeled instances. This technique is useful for achieving high-accuracy pseudo-labeling but its computational complexity makes it impractical for practical use.

*Zhu, Yu & Jing (2013)* applied Semi-Supervised Learning approach for text representing and term classification based on term-weight in *Zhu, Yu & Jing (2013)*. The experimental results proved the effectiveness of results by the proposed method when compared to the results of supervised classification methods.

More recently, *Li et al. (2019)* presented an incremental SSL method for classification of streaming data in *Li et al. (2019)*. This approach proposes a model consisting of generative network used to learn representations from input (autoencoders), discriminant structure used to regularize the generative network by building pairwise similarity/dissimilarity (semi-supervised hashing), and the bridge which connects the generative network with the discriminant structure. The proposed approach employs transductive learning and falls in the category of generative methods of semi-supervised learning. They compared their incremental model on evolving streaming data with the state-of-the-art incremental learning approaches like Learn++, AdalinMLP, etc. This approach named ISLSD/ISLSD-E showed to be experimentally more accurate than the supervised incremental learning approaches in competition. Despite its good performance, the proposed approach doesn't provide a comprehensible rule-based classifier.

*Wang et al. (2021)* presented an ensemble framework named Ensemble of Auto-Encoding Transformation (EnAET) for self-training of images in *Wang et al. (2021)*. They employed both the spatial and non-spatial transformations for training the deep learning neural network for both the labeled and unlabeled data. This technique outperformed other state-of-the-art self-training methods in experiments. EnAET is neither a rule-based system nor is it used for discrete data.

As per our knowledge, there exists no associative classification approach for self-training, self-labeling or even entire semi-supervised classification.

We argue that since associative classification increases the robustness and confidence of classification rules (*Shahzad & Baig, 2011*; *Hadi, Al-Radaideh & Alhawari, 2018*; *Venturini, Baralis & Garza, 2018*), it is more logical and a natural way to incorporate associative classification for pseudo-labeling and rule construction in self-trained semi-supervised classification. Thus the main contribution of the proposed approach is the utilization of ACO-based associative classification for self-training and construction of comprehensible rule-based classifier to achieve higher classification accuracy than self-trained versions of classical classification algorithms.

## PROPOSED METHODOLOGY

The proposed approach consists of three components, the transductive self-training mechanism of SSL, principles of associative classification and rule construction by ACO.

Algorithm 1 illustrates the proposed ST-AC-ACO algorithm. The algorithm starts by applying pre-processing (if necessary). If the dataset ($D$) is not in nominal (categorical) form, it is discretized (line 1). If $D$ contains any class(es) with too few instances in $D$ to constitute a pattern, such instances are considered outliers. These instances are either to be merged with any closely-related class instances or removed from $D$ (line 2) if required. This concludes the pre-processing.

The labeled dataset ($L$), the unlabeled dataset ($U$) and the test set ($TestSet$) are initialized from the input dataset $D$ (line 3). The extended labeled set ($EL$) is initialized with $L$. The $EL$ acts as training set in the algorithm.

The *While* loop (lines 6–22) executes until all the instances in $U$ have been pseudo-labeled and moved to $EL$.

Pheromone is initialized four as illustrated in Eq. (9):

$$\tau_{ij} = \frac{1}{|Terms|} \tag{9}$$

where $Terms$ is the set of terms in the data set and $\tau_{ij}$ is the pheromone value for the *edge* from node $i$ to $j$. A terms is represented by an edge in of a graph (see Fig. 2).

The heuristic function (line 9) is the second component for probabilistic selection of terms. Eq. (10) is used to calculate heuristic value for the selection of the first term.

$$\eta_i = \frac{|term_i, class_k| + 1}{|term_i| + |classes|} \tag{10}$$

where $\eta_i$ is the *heuristic value* for selection of the $i$th term as the 1st term of the rule

---

**Algorithm 1 ST-AC-ACO.**

1: Discretize Dataset $D$

2: Remove outliers from $D$

3: Read $L$, $U$, $TestSet$ from $D$

4: Set $EL \leftarrow L$

5: Initialize $minSupp$, $minConf$, $No\ of\ Ants$

6: **while** $U \neq \phi$ **do**

7:     Set $RuleList \leftarrow \phi$

8:     Initialize $phermone$

9:     Initialize $heuristic$

10:     **for** Each class label $c$ **do**

11:         Set $Class\_Rules \leftarrow \phi$

12:         Set $TRules \leftarrow ConstructTermRules()$

13:         Calculate Selection Probabilities See Eq. (8)

14:         Set $ARules \leftarrow ConstructAntRules()$

15:         Set $Class\_Rules = TRules \cup ARules$

16:         Set $RuleList \leftarrow RuleList \cup Class\_Rules$

17:     **end for**

18:     Sort $RuleList$ by $confidence$ and $support$(in descending order).

19:     Randomly select $Instances$ from $U$.

20:     Assign class labels to each instance in $Instances$ using $RuleList$.

21:     Set $U \leftarrow U - Instances$ and $EL \leftarrow EL \cup Instances$.

22: **end while**

23: Prune $RuleList$ and remove duplicate rules (if any).

24: Test $RuleList$ on $TestSet$.

25: Display Results.

---

antecedent while $class_k$ represents $k$th class. The expression $|term_i, class_k|$ in the numerator of the fraction denotes the number of instances in $EL$ which contain $term_i$ with class label $k$. The denominator is the sum of the total number of instances containing the $term_i$ in $EL$ and total number of classes. This heuristic is directly used as the selection probability of the term $i$.

After the selection of the first term, heuristic function for the each subsequent term is calculated by Eq. (11). This equation is used for calculation of the selection probability (see Eq. (8)) of term $j$ given that term $i$ is currently selected.

$$\eta_{ij} = \frac{|term_i, term_j, class_k| \times |term_j, class_k|}{|term_i, class_k| \times |class_k|} \tag{11}$$

where $\eta_{ij}$ is the heuristic value for link between the current item $term_i$ and a selection candidate item $term_j$ while $|term_i, term_j, class_k|$ represents the number (frequency) of

instances containing itemset {*item_i*, *item_j*, *class_k*} *i.e.*, instances in which *term_i*, *term_j* and *class_k* occur in the same instance.

The algorithm discovers associative classification rules for each class one by one. The lines 10–17 construct class rules for each class *c*. Since there can exist non-associative classification rules consisting of one term, *TRules* set (line 12) would contain single-term rules constructed by the Algorithm 5.

Selection probability (line 13) is used to guide ants for selection of the terms. *ARules* is the list of rules constructed by ants (line 14) returned by the function *ConstructAntRules()* which is demonstrated in Algorithm 3. *Class_Rules* list is constructed by the union of *TRules* and *ARules* (line 15). *Class_Rules* are in turn added to the global rule list named *Rules* (line 16).

After the rules for all classes are constructed, the *RulesList* is sorted (line 18) in the descending order of *confidence* and then *support* (if two rules have equal value of *confidence*).

The process of randomly selection of unlabeled instances from *U* and assigning them the most suitable labels has been described in lines 19 and 20. The selected instances are called *pseudo-labeled* and are moved from *U* to *EL* set (line 21). Number of instances added from *U* to *EL* is illustrated in Eq. (12)).

Pseudo-labeling of a chosen instances is done using sorted rules by confidence in descending order. An instance covered by a rule with a higher confidence is more likely to be pseudo-labeled correctly. Items of each selected instance are compared to respective terms of the antecedents of the sorted rules. The consequent of the *first rule* whose antecedent matches an instance is assigned as the label of the instance. Moreover, there are *Support* and *Confidence* values associated to every pseudo-labeled instances. The *Support* and *Confidence* of the covering rule are assigned to the *Support* and *Confidence* fields of each covered pseudo-labeled instance.

The pseudo-labeling by associative classification is expected to be more accurate than non-associative classification rules because of associativity among terms of the dataset and between the set of associative terms and the class labels.

$$n = \left( \begin{array}{ll} |U|, & \text{if } \mu >= |U| \\ r, & \text{otherwise.} \end{array} \right. \tag{12}$$

where *n* represents the number of instances to be selected from *U*, *mu* is the user-defined parameter which sets the maximum number of instances to be selected in one iteration, *U* represents the set of unlabeled instances and *r* is a random number [1, *μ*]. Moreover, the instances are chosen randomly from *U* to move to *EL*. This mechanism provides some level of dynamic extension of the *EL* as opposed to existing approaches like the approaches proposed in *Jiang, Zhang & Zeng (2013)*, *Triguero, Garca & Herrera (2015)*, etc, which employ the mechanism of selecting, pseudo-labeling and adding (to the *EL* set) a fixed static number of instances from *U* set.

The constructed rules are then pruned to remove any redundant terms from rules (line 23) and then duplicate rules are removed if there exist any. Finally the *RuleList* is used to calculate the accuracy on *TestSet* and report the results (lines 24–25).

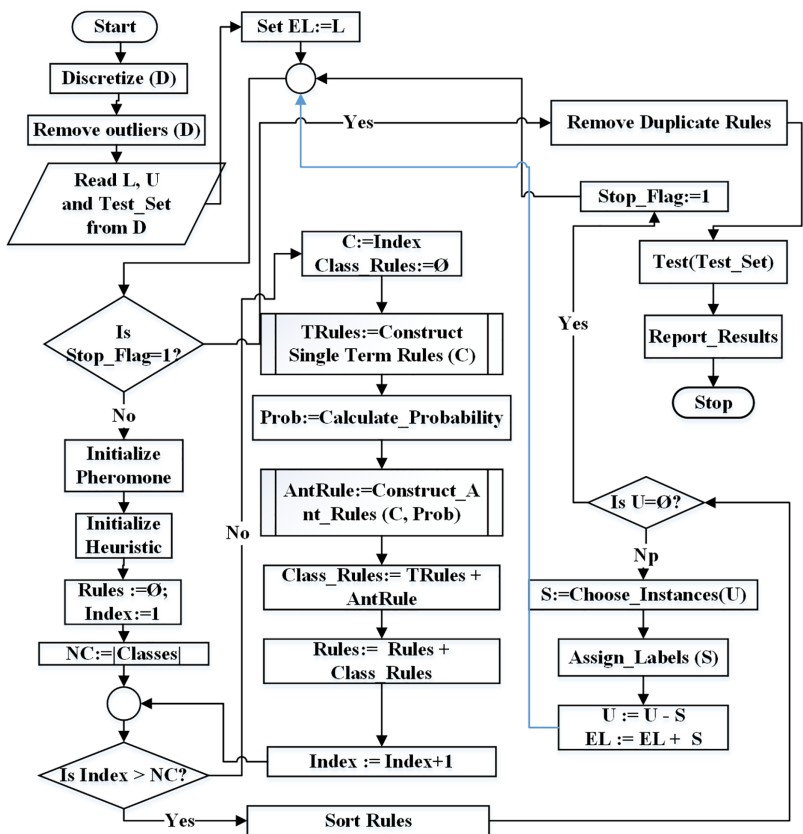

**Figure 4 Flowchart for the proposed ST-AC-ACO algorithm.**

Figure 4 represents the flowchart of the proposed technique. The identifiers *NC* and *Index* represent the number of classes and current class index respectively. Similarly, *Rules* represents the rule list, *L* represents the set of labeled instances, *U* represents the set of unlabeled instances while *EL* represents the set of extended labeled instances.

Algorithm 2 illustrates the process of construction of single-term rules. Such rules determine the association of each individual term of the dataset to class labels. Line 2 describes the construction of single-term rule for each term in the dataset exhaustively for class *c*. Line 3 describes the calculations of support (Eq. (14)) and confidence (Eq. (15)) of the single-term rule. Line 5 sets *pheromone* trails to 0 from the *term* of the current rule if *support* is less then the user-defined *minSupport* threshold. If *support* and *confidence* values of the current rule meet the *minSupport* and *minConfidence* thresholds respectively, the *rule* is added to the *Rules* list (line 7). If the *support* of a rule meets the *minSupport* threshold but the *confidence* doesn't meet the *minConfidence*, neither the rule is added to the rule list, nor the pheromone is modified.

Calculation of support and confidence measures of an associative rule is the most critical step in evaluation of the constructed rule. Although, calculations of these measures have been described in Background (see Definitions 6 and 7), but they are useful in case of supervised learning where frequency of terms is simply their count. But in the case of

---

**Algorithm 2** ConstructTermRules(c).

1: **for** Each *term* **do**   Rule for each term
2:    Construct 1-term rule for term, such that (term => c).
3:    Calculate *support* and *confidence* of *rule*.
4:    **if** *support < minSupp* **then**
5:       Set *pheromone ← 0*: for all *terms* trails.
6:    **else if** *support ≥ minSupp* And *con f indence ≥ minCon f* **then**
7:       Set *Rules ← Rules ∪ rule.*
8:    **end if**
9:    Return *Rules*
10: **end for**

---

**Algorithm 3** ConstructAntRules(c, Prob).

1: Set *g ← 2.*Generation counter
2: **while** *g ≠ |attributes|* And *coverage ≤ minCoverage* **do**
3:    Set *MultiRules ← φ*Multi-term rules
4:    Set *t ← 1* Ant index Ant index
5:    **repeat**
6:       Let ant *t* construct a maximum of *g*-term *rule* such that (*rule => c*) using selection probability *prob*.
7:       Set *t ← t + 1*
8:    **until** *t > noO f Ants*
9:    **for** Each *rule* constructed by ants **do**
10:       Calculate *support* and *confidence* of rule
11:       **if** *support ≥ minSupp* And *≥ minConf* **then**
12:          Set *MultiRules ← MultiRule ∪ rule*
13:       **end if**
14:    **end for**
15:    Set *coverage ← find_coverage(multiRules).*
16:    Update *pheromone.*
17:    Set *g ← g + 1*
18: **end while**
19: return *MultiRules*

---

semi-supervised learning, the pseudo-labeled instances can't be guaranteed to have correct labels, therefore, a pseudo-labeled instance should not have its weight equal to a labeled instance.

A notable contribution of the proposed approach is to define the weight for pseudo-labeled instances for calculation of support and confidence measures of a rule that covers them. A labeled instance has a weight of one for calculating the support and confidence of

the rule covering the instance. But this is not the case with a pseudo-labeled instance. A pseudo-labeled instance has associated values of *Support* and *Confidence* which are assigned from the respective values of the rule through which the instance was pseudo-labeled.

The modified support calculation is presented in Eq. (13).

$$Support = \frac{Support_l \times Frequency_l^c + Support_p \times Frequency_p^c}{Frequency_l^c + Frequency_p^c} \tag{13}$$

where $Support_l$ denotes the support of the rule in labeled instances calculated using Eq. (3). $Frequency_l^c$ is the number of labeled instances containing the class label $c$. The superscript variable $c$ denotes the consequent (class label) of the rule whose support is being calculated. Similarly, $Support_p$ represents the rule support in pseudo-labeled instances while $Frequency_p^c$ denotes the frequency of pseudo-labeled instances with rule class label $c$.

When an unlabeled instance is pseudo-labeled, its *Support* and *Confidence* fields are assigned respective *support* and *Confidence* values of the rule through which the instance was assigned the class label. Thus when such an instance is part of *EL* and another rule covers the instance, the instance is not counted. Instead its *Support* is added while calculating the support of the covering rule. The sum of *support* of pseudo-labeled instances covered by the rule is divided by the frequency of the pseudo-labeled instances having the rule class $c$ as expressed in Eq. (14).

$$Support_p = \frac{\sum_{i=1}^{Frequency_p^c} Support_i}{Frequency_p^c} \tag{14}$$

where $Frequency_p^c$ denotes the number of pseudo-labeled instances in the rule class $c$ and $Support_i$ represents the support value of the pseudo-labeled instance $i$. Obviously, the sum of *Support* of pseudo-labeled instances is less than frequency of these instances as the frequency of each such instance is 1 while $0 < Support_i \leq 1$. The same is the case with the confidence. It is the ratio of the sum of *Confidence* values of pseudo-labeled instances having class label $c$ and are covered by the rule to the frequency of instances covered by the rule (antecedent) independent of class. Eq. (15) is used to calculate the $Confidence_p$ of the rule covering pseudo-labeled instances.

$$Confidence = \frac{SupportCount_l \times Confidence_l + SupportCount_p \times Confidence_p}{SupportCount_l + SupportCount_p} \tag{15}$$

where $SupportCount_l$ and $SupportCount_p$ denote the number of cases (instances) covered by the antecedent of the rule in the labeled and pseudo-labeled instances respectively. $Confidence_l$ is the rule confidence in the labeled instances and calculated using the Eq. (4), while $Confidence_p$ is the rule confidence in the pseudo-labeled instances. It is calculated using the Eq. (16).

$$Confidence_p = \frac{\sum_{i=1}^{Frequency_r^c} Confidence_i}{Frequency_r} \tag{16}$$

where $Frequency^c_r$ represents the number of pseudo-labeled-instances matching with both the antecedent and the consequent of the rule whose confidence is being calculated, $Confidence_i$ denotes the confidence value associated with the instance $i$ while $Frequency_r$ is the frequency of pseudo-labeled instances covered by the rule independent of class.

Algorithm 3 illustrates the construction of associative classification rules by ants. Variable $g$ represents the *generation index* of the ant rules. Each ant constructs an associative classification rule consisting of $g$ number of terms in its antecedents. The initial value of $g$ is set to 2 (line 1). The *while* construct (lines 2–18) present the evolutionary process of the rule construction. The variable *minCoverage* is a user-defined parameter which specifies the proportion of *EL* that has to be covered by the *MultiRules* rule list constructed by ants before termination of the rule construction process and its value is in range [0, 1]. Lines 5–8 describe how each ant $t$ constructs a rule consisting of at most $g$ terms. The variable $c$ represents the selected class. For each rule in the ant-constructed rules, if *support* and *confidence* meet the threshold values, the rule is added to the *MultiRules* list (lines 9–14). During construction of a multi-term rule, there are two steps involved. In the first step, an ant has to select first term using Eq. (10).

The second step is to select subsequent terms of a multi-term rule. The pheromone (definition 8) for each possible ant path and heuristic function (definition 9) are the component of the calculation of the selection probability of each subsequent term (definition 10, Eq. (8)). Every subsequent term is probabilistically selected and added to the rule of the current ant $t$.

The *pheromone* and consequently the probability matrices are updated after all ants of the $g$-th generation construct their rules. The pheromone for each path from $term_i$ to $term_j$ is evaporated and is updated using Eq. (17).

$$\tau_{ij}(g+1) = \tau_{ij}(g) \times (1 - \rho) \tag{17}$$

where $g$ represents generation (or iteration) while $\rho$ is a user-defined parameter called pheromone evaporation rate (definition 11).

The *coverage* on instances in class $c$ by *multiRules* set is calculated after each generation $g$ (line 15). If *coverage* meets the *minCoverage* threshold, the *While* loop of line 2 is terminated.

The pheromone of paths used in construction of rules that were added to the *MutiRules* list is updated (line 1) using Eq. (18).

$$\tau_{ij}(g+1) = \tau_{ij}(g) + \tau_{ij}(g) \times \left(1 - \frac{1}{1 + conf_r}\right) \tag{18}$$

where $r$ represents index of the rule in *Rules* list. The higher the confidence of a rule implies the higher the value of the appropriate pheromone trail.

The computational complexity of the proposed algorithm (Algorithm 1) needs to be calculated in two phases. In the first phase, the computational cost of training process is calculated (lines 7–18). The second step is to find the computational cost of pseudo-labeling and re-training.

The training of associative classifier consists of pheromone and heuristic initialization and rule construction. If there are $r$ number of terms, the size of each of the pheromone and the heuristic matrixes is $r^2$. Thus time complexity of initialization of pheromone and heuristic becomes $O(r^2)$. Rule construction is the most complex part of the training step. Construction of single-term rule (line 12) is performed by calling Algorithm 2. A rule for each term $r$ is constructed with the time complexity of $O(r)$. Let the number of instances in the training set be $n$. The time complexity of calculation of support and confidence for $t.A^2$ rules becomes $O(r.n)$. But since this process is repeated for every class $c$ in the training set ($EL$), the time complexity of single-rule construction becomes $O(c.r.n)$

Multi-term (ant) rules are constructed (line 14) by calling Algorithm 3. The *While* loop of that algorithm (lines 2–18 for at most $|attributes| - 1$ times. Let $A$ represent the number of attributes. Each ant $t$ constructs a rule of at most $g$ conditions (in rule antecedent) (lines 5–8) by selecting one term at a time. Moreover, $g$ can reach at most to $A$. Thus the worst-time complexity of rule construction by ants is $O(t.A^2)$.

Let the number of instances in the training set be $n$. The time complexity of calculation of support and confidence (line 10) for $t.A^2$ rules becomes $O(t.A^2.n)$.

There are $r$ terms and each of them in the training set. Each term is normalized in range [0, 1]. The pheromone of update is performed at most $T.A^2$ rimes. Thus the time complexity for pheromone update is $O(t.A^2.r)$.

Thus the run time of Algorithm 6 is $O(t.A^2) + O(t.A^2.n) + O(t.A^2.r)$. Sine $n$ (number of instances) is expected to be much larger than $r$ (number of terms), therefore, the time complexity becomes $O(t.A^2.n)$.

Algorithm 3 is called $c$ (number of classes) times, the time complexity of construction of ant rules becomes $O(c.t.A^2)$. Since this complexity is higher than time complexity of single-term rule construction, therefore, this is also the worst runtime complexity of the training phase.

The training is repeated after pseudo-labeling of unlabeled instances and adding them to the training set ($EL$). The number of instances selected in each iteration of the *While* loop (lines 6–22) of Algorithm 1 is random in range $[1, \mu]$ where $mu$ is a user-defined integer value indicating the maximum number of instances that may be chosen for pseudo-labeling in one iteration. The number of chosen instances is 1 in each iteration in the worst case. As the instances from unlabeled set $U$ move to extended labeled set $EL$, the size of $U$ shrinks but that of $EL$ increases. The number of total instances in the data set is $n$ which is the sum of the number of labeled instances and the number of labeled (+ pseudo-labeled) instances and remains constant. The training phase is repeated $n$ times during pseudo-labeling. Hence the time complexity of the proposed algorithm is $O(n.c.t.A^2.n)$ which can be written as $O(c.t.A^2.n^2)$ where the number of instances of the underlying dataset $n$ is the major factor.

## EXPERIMENTAL RESULTS

For the purpose of evaluation of performance of the proposed ST-AC-ACO algorithm and comparison with other self-training techniques, 25 datasets from KEEL dataset repository were used (https://sci2s.ugr.es/keel/semisupervised.php). These datasets include some

**Table 5 Datasets used for experiments.**

| Sr. No. | Dataset | \|Att\| | \|Ins\| | \|Class\| | Class Dist |
|---|---|---|---|---|---|
| 1 | Appendicitis | 7 | 106 | 2 | Imbalanced |
| 2 | Australian | 14 | 690 | 2 | Balanced |
| 3 | Automobile | 24 | 159 | 4 | Balanced (Pre) |
| 4 | Banana | 2 | 5,300 | 2 | Balanced |
| 5 | Breast Cancer | 9 | 286 | 2 | Imbalanced |
| 6 | Chess | 36 | 3,196 | 2 | Balanced |
| 7 | Cleveland | 13 | 297 | 2 | Balanced (Pre) |
| 8 | Contraceptive | 9 | 1,473 | 3 | Balanced |
| 9 | CRX | 15 | 653 | 2 | Balanced |
| 10 | Flare | 11 | 1,066 | 6 | Imbalanced |
| 11 | German | 20 | 1,000 | 2 | Imbalanced |
| 12 | Glass | 9 | 214 | 3 | Balanced |
| 13 | Haberman | 3 | 306 | 2 | Imbalanced |
| 14 | Heart | 13 | 270 | 2 | Balanced |
| 15 | Iris | 4 | 151 | 3 | Balanced |
| 16 | LED7Ligit | 7 | 550 | 10 | Balanced |
| 17 | Lymphography | 18 | 148 | 2 | Balanced (Pre) |
| 18 | Magic | 10 | 19,020 | 2 | Imbalanced |
| 19 | Mammographic | 5 | 830 | 2 | Balanced |
| 20 | Mushroom | 22 | 5,644 | 2 | Balanced |
| 21 | Nursery | 9 | 12,630 | 3 | Balanced (Pre) |
| 22 | Pima | 8 | 760 | 2 | Balanced |
| 23 | Saheart | 9 | 462 | 2 | Balanced |
| 24 | Tae | 5 | 151 | 3 | Balanced |
| 25 | Titanic | 3 | 2,201 | 2 | Imbalanced |

reasonably large datasets like *Banana* (5,300 instances), *Chess* (3,196 instances), *Magic* (19,020 instances), *Mushroom* (5,644 instances), *Nursery* (12,630 instances) and *Titanic* (2,201 instances).

For the purpose of performance comparison, 5 top-performing state-of-the-art self-training techniques were chosen for competition with the proposed ST-AC0ACO technique. Competing techniques include ST-C4.5, ST-Naive Bayes (ST-NB) (*Yarowsky, 1995b*), Sequential Minimal Optimization (ST-SMO) (*Kumar et al., 2020*) which is an implementation of Support Vector Machine (SVM), Self Training with Editing (SETRED) (*Li & Zhou, 2005*) and Ant-Based Semi-Supervised Classification (APSSC) (*Halder, Ghosh & Ghosh, 2010*).

Table 5 displays the datasets used to evaluate the performance of the proposed approach and other self-training approaches. The column with heading |*Att*| represents the number of attributes of datasets, |*Inst*| represents the number of instances of datasets, |*Class*| represents the number of classes of data datasets and the last column demonstrates whether a dataset is either balanced or imbalanced with respect to class distribution.

It is important to note that the training and test sets are prepared using uniform class distribution. Instances from training set are randomly picked from each class according to the uniform class distribution to remove class labels before adding to *U*. The remaining instances are added to *L*. The key step is to maintain the specific proportion of labeled instances in *L* from the training set. Further detail has been explained in Experimental Results.

## Pre-processing

Majority of the datasets used in the evaluation consist of balanced class distribution. Datasets mentioned as *Balanced Pre* in Table 5 were pre-processed to merge instances of low-frequency classes into new higher-frequency class instances for creating maintaining a balance in the class distribution of such datasets. For instance, the dataset *Automobile* contains instances of six class labels, three of which make up about 23% instances of the dataset. Those three classes were merged into a single class to create a balanced dataset. This pre-processing helped only in datasets where low-frequency class instances collectively became sufficient to form a frequency close to that of all other classes. But in some cases instances with very low-frequency class labels were still too far from creating a balanced class distribution after merging. For instance, *Nursery* dataset originally consists of instances of five classes, two of which have only about 2.5% representation in the entire dataset while rest of the classes have almost equal frequency distribution. Thus instances of such classes were considered as *noise* and were removed from the dataset and the dataset was left with three class labels. To perform this pre-processing, the *Data Filter* feature was used. The original and pre-processed versions of such datasets are publicly available at *Awan (2020)*.

Another important pre-processing task was to *discretize* the continuous data because the proposed algorithm and competing self-training algorithms run on discrete values. For this purpose, the *discretize* filter (with default options) of *Weka Machine Learning Workbench 3.7* was used (*Benchmark, 2021*). Figure 5 displays a screenshot of the Discrete filter of Weka 3.7 used for discretization process.

## Experimentation setup

Table 6 lists parameter values used in training phase of the ST-AC-ACO and competing state-of-the-art self-training classification algorithms. Number of ants, pheromone evaporation rate ($\rho$) and minimum coverage (*MinCoverage*) have been set as in *Shahzad & Baig (2011)*, while values for minimum support and minimum confidence threshold have been specified by determining the most suitable values through experimentation. Minimum coverage value 1.0 means that training will stop when all instances of the *EL* have been covered by the list of discovered rules. Parameters for ST-C4.5, ST-SMOG (SVM), SETRED and APSSC have been set according to the setting in *Zhu, Yu & Jing (2013)*. Self-Training C4.5 (ST-C4.5) requires two parameters namely confidence level *c* and minimum number of itemsets per leaf of the decision tree. The algorithm post-prunes the tree. Self-training Sequential Minimal Optimization (ST-SMO) is SVM variant (*Kumar et al., 2020*). Parameter *C* is set to value 1 to achieve higher training accuracy

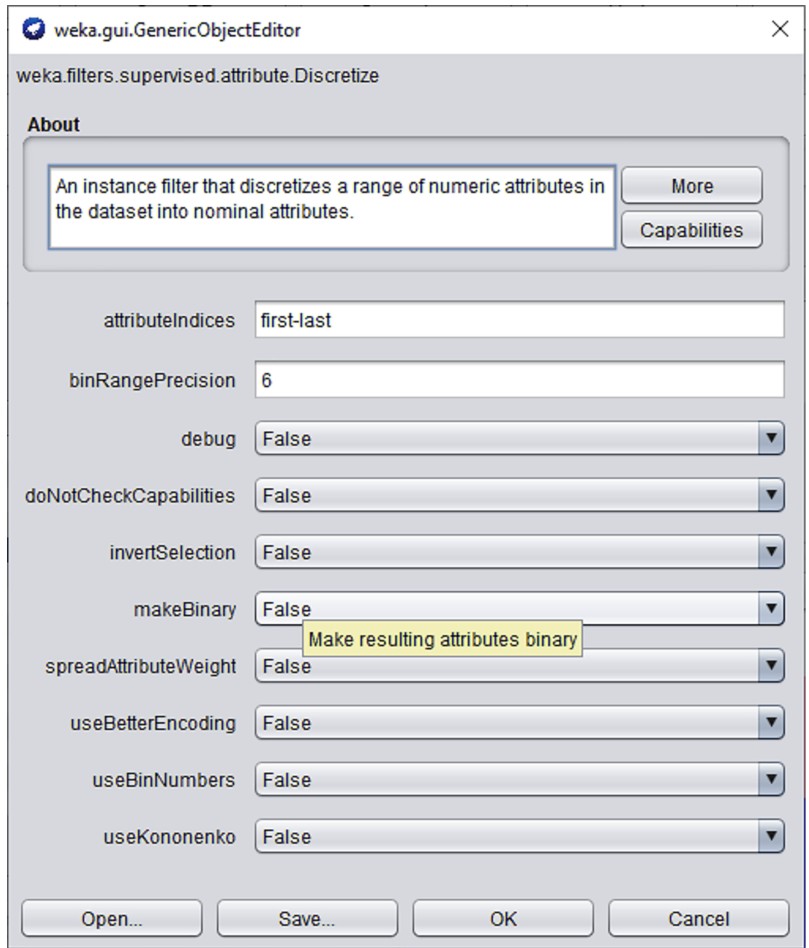

**Figure 5 Weka 3.7 Discretize filter with default options.**

because the ST-SMO is trained on labeled data to correctly assign labels to unlabeled instances during training. The selected three competitors have been the best performing self-training algorithms in the KEEL tool (*Zhu, Yu & Jing, 2013*). That is why they have been chosen for comparison with the performance of he proposed ST-AC-ACO algorithm. SETRED uses the amending process to continuously edit the pseudo-labeling of the *EL* set. APSSC is an Ant-based semi-supervised classification approach that does not involve exploiting associativity among dataset elemis. This algorithm does not require number of ants parameter as this value is set dynamically to the number of classes in the dataset during execution of the algorithm. However pheromone evaporation rate $\rho$ is set quiet high because number of ants is much smaller in most of the cases.

The proposed ST-AC-ACO algorithm has been implemented in C# while its competitor algorithms used in experimentation have been part of the Semi-Supervised Learning module of the KEEL (*Alcal Alcalá-Fdez et al., 2009*) software. A significant difference between implementation of ST-AC-ACO and KEEL implementation is that ST-AC-ACO implementation does not require separate partition files for each partition of datasets. The software is developed to create partition during runtime and to remove labels of the

**Table 6 Algorithm parameters used in experiments.**

| Algorithm | Parameter name | Value |
|---|---|---|
| ST-AC-ACO | No of ants | 30 |
| | Min support | 0.05 |
| | Min confidence | 0.45 |
| | $\rho$ | 0.09 |
| | Min coverage | 1 |
| ST-C4.5 | c | 0.25 |
| | i | 2 |
| | Pruning | Post-prune |
| ST-NB | None | N/A |
| ST-SMO | Kernal type | Polynomial |
| | Polynomial degree | 1 |
| | Fit logistic model | TRUE |
| | C | 1 |
| | Tolerance parameter | 0.001 |
| | $\varepsilon$ | $1.00E-12$ |
| SETRED | Max iterations | 40 |
| | Threshold | 0.1 |
| APSSC | Spread of the Gaussian | 0.3 |
| | Confidence | 0.7 |
| | $\rho$ | 0.7 |

instances of the unlabeled instances before training. Thus the user doesn't have to prepare labeled partitions for datasets. The implementation software for ST-AC-ACO and pre-processed datasets can be found online (http://www.hamidawan.com.pk/research/).

The training of ST-AC-ACO consists of two phases, the training on labeled data phase and the pseudo-labeling phase. The algorithm works on discrete data. The test data is kept separate from the training set. The training data is then partitioned into labeled and unlabeled data according to desired percentage of labeled data. For instance, consider *German* dataset which contains 1,000 instances. In 10-fold cross-validation, 10% (100 instances) of the dataset become test set in each fold, while the rest (900 instances) will make up the training set. Assuming that the labeled proportion is 20%, thus labeled set (*L*) and the extended labeled set (*EL*) will contain 180 instances (20% of the training set) while the unlabeled set *U* will consist of the remaining 720 instances. The classifier will first discover associative classification rules as discussed in Proposed Methodology. The model constructs rules for each class by choosing one class at a time. Single-term rules for each term in the dataset are discovered (Algorithm 2) and added to the rule list. Then ACO stochastic search mechanism is used to construct associative rules (Algorithm 3). Rules are stored in a global rule list. After training is complete, the pseudo labeling phase starts. A small amount of instances is chosen from *U* and presented to (sorted by confidence) rule list and most class label is assigned to each instance.

The model is retrained until all instance from the unlabeled set *U* have been modes to *EL*. Finally testing for the fold is performed and results are reported.

## Performance evaluation

A total of 10-fold cross-validation mechanism for evaluation and comparison is used during experimentation where 90% data is used for training and 10% data is used for testing in each fold. Labeled and unlabeled partitions are made from the training data. A total of Two performance measures were used for comparison, *i.e.*, classification accuracy and *Cohen's Kappa statistic* (K statistic) which is an alternative measure of $F_1$ measure (*Ben-David, 2007*). K statistic is the measure of agreement between the actual values of classes with their predicted values by the classifier. Thus, like, *precision* and *recall* measures, which are components of the $F_1$ measure, the K statistic operates on the confusion matrix of the classification results. The distinctive feature of K statistic is that it provides a scalar value for multi-class confusion matrix. According to its nature, K statistic penalizes class predictions based on merely higher frequency of a majority class. This feature makes K-statistic more suitable for performance analysis and validation of semi-supervised classification techniques (*Triguero, Garca & Herrera, 2015*).

The experimentation was setup for 4 sets consisting of 10%, 20%, 30% and 40% labeled data. Tables 7 to Table 10 demonstrate the comparison of performance the classification accuracy comparison of the above-mentioned algorithms respectively. The Figs. 6–9 present the visualization of the appropriate tables mentioned above.

As obvious from Table 7, ST-AC-ACO algorithm comprehensively beat its competing algorithms on *Appendicitis* (with 89.64% accuracy as compared to 80.25% accuracy of ST-C4.5 algorithm), *Automobile* (with 54.25% accuracy as compared to 43.38% accuracy of SETRED algorithm), *Breast cancer* (with 78.34% accuracy as compared to 72.42% accuracy of ST-Naive Bayesian algorithm), *Cleveland* (with 67.03% accuracy as compared to 53.39% accuracy of ST-NB), *Glass* (with 61.13% accuracy as compared to 54.02% accuracy of SETRED), *Heart* (with 89.26% accuracy as compared to 77.78% accuracy of APSSC), *Mammographic* (with 98.07% accuracy as compared to 80.22% accuracy of APSSC), *Nursery* (with 87.95% accuracy as compared to 77.04% accuracy of ST-C4.5), *Pima* (with 81.25% accuracy as compared to 69.00% accuracy of ST-NB), *Sahrart* (with 74.03% accuracy as compared to 65.59% accuracy of APSSC), *Tae* (with 58.29% accuracy as compared to 41.08% accuracy of SETRED) *Titanic* (with 83.69% accuracy as compared to 77.56% accuracy of APSSC). Moreover, ST-AC-ACO beat all other algorithms on the largest selected *Magic* dataset by a small margin and showed 100% accuracy on *Mushroom* dataset. With the help of Wilcoxon's signed rank test *Garca et al. (2010)*, it is shown that ST-AC-ACO beat non-associative self-training versions of classification algorithms in 10 of 25 datasets with a significant margin on 10% labeled data.

Table 8 presents accuracy comparison of the self-training algorithms on 20% labeled data. ST-C4.5 came closer to ST-AC-ACO over *German* dataset by showing comparable accuracy 69.18% to ST-AC-ACO's 70.20%). ST-NB showed comparable accuracy (89.00%) on *Appendicitis* to that of ST-AC-ACO (87.64%), while it was beaten by ST-AC-ACO on 10% labeled *Appendicitis* dataset. ST-NB beat ST-AC-ACO by showing

**Table 7 Classification comparison on 10% labeled data.**

| Datasets | ST-AC-ACO (%) | ST-C4.5 (%) | ST-NB (%) | ST-SMO (%) | SETRED (%) | APSSC (%) |
|---|---|---|---|---|---|---|
| Appendicitis | 89.64 | 80.25 | 79.45 | 79.15 | 73.73 | 67.73 |
| Australian | 85.80 | 81.93 | 75.83 | 80.02 | 80.43 | 83.77 |
| Automobile | 54.25 | 37.89 | 34.67 | 29.52 | 43.88 | 43.12 |
| Banana | 80.89 | 80.22 | 58.57 | 84.54 | 86.38 | 82.40 |
| Breast Cancer | 78.34 | 69.66 | 72.42 | 69.89 | 68.35 | 67.24 |
| Chess | 77.16 | 95.43 | 80.10 | 89.64 | 81.04 | 83.26 |
| Cleveland | 67.03 | 51.06 | 53.39 | 41.84 | 52.94 | 48.58 |
| Contraceptive | 71.21 | 47.33 | 74.12 | 79.88 | 41.48 | 42.29 |
| CRX | 87.58 | 86.00 | 75.68 | 82.26 | 81.11 | 84.63 |
| Flare | 71.49 | 71.57 | 71.12 | 51.24 | 64.45 | 52.35 |
| German | 73.30 | 68.68 | 67.81 | 59.02 | 66.60 | 62.10 |
| Glass | 61.13 | 49.66 | 40.94 | 48.93 | 54.02 | 38.01 |
| Haberman | 75.80 | 70.21 | 79.69 | 61.88 | 62.11 | 58.43 |
| Heart | 89.26 | 72.33 | 69.59 | 76.26 | 74.44 | 77.78 |
| Iris | 93.33 | 81.48 | 79.26 | 94.18 | 91.33 | 94.67 |
| LED7Ligit | 65.00 | 60.74 | 56.10 | 56.81 | 61.80 | 69.40 |
| Lymphography | 54.73 | 62.32 | 5.59 | 54.22 | 68.33 | 65.52 |
| Magic | 77.15 | 72.17 | 63.18 | 73.94 | 68.40 | 63.79 |
| Mammographic | 98.07 | 79.39 | 73.30 | 77.10 | 75.80 | 80.22 |
| Mushroom | 100.00 | 99.55 | 92.43 | 99.39 | 99.45 | 97.55 |
| Nursery | 87.95 | 77.04 | 76.88 | 58.48 | 77.01 | 64.83 |
| Pima | 81.25 | 66.10 | 69.00 | 62.07 | 65.65 | 66.83 |
| Saheart | 74.03 | 63.82 | 64.78 | 62.27 | 63.00 | 65.59 |
| Tae | 58.29 | 38.97 | 36.61 | 40.76 | 41.08 | 40.29 |
| Titanic | 83.69 | 75.51 | 71.06 | 70.60 | 64.02 | 77.56 |

81.92% in comparison of ST-AC-ACO's 74.49% on *Haberman* dataset. ST-SMO beat ST-AC-ACO on *Contraceptive* dataset by showing 84.05%accuracy against 74.54% of ST-AC-ACO. ST-ACO attained a comprehensive lead on *CRX* dataset by showing 96.94% accuracy in comparison to 86.28% accuracy of APSSC. Results on rest of the datasets remained almost unchanged as far as ST-AC-ACO's performance is concerned. Wilcoxon's tests show that despite of being behind on a couple of occasions, ST-AC-ACO beat all of its competitors in accuracy on 9 out of 25 datasets by a significant margin. Figure 9 demonstrates the visual analysis of the results for the results displayed in Table 8.

Table 9 displays summary of accuracy comparison of self-training algorithms on 30% labeled data. Figure 8 presents the visual analysis of the same results. ST-AC-ACO showed comparable results to all competitor techniques on *Banana* dataset for the first time and attained classification accuracy of 87.80%. Similarly, AC-ACO showed much improved results on *Chess* dataset to majority of competitors by attaining 96.37% accuracy. Moreover, ST-AC-ACO attained comprehensive lead over all of its competitors on *Breast Cancer* (with accuracy of 77.25%), *Contraceptive* (with accuracy of 73.12%) and

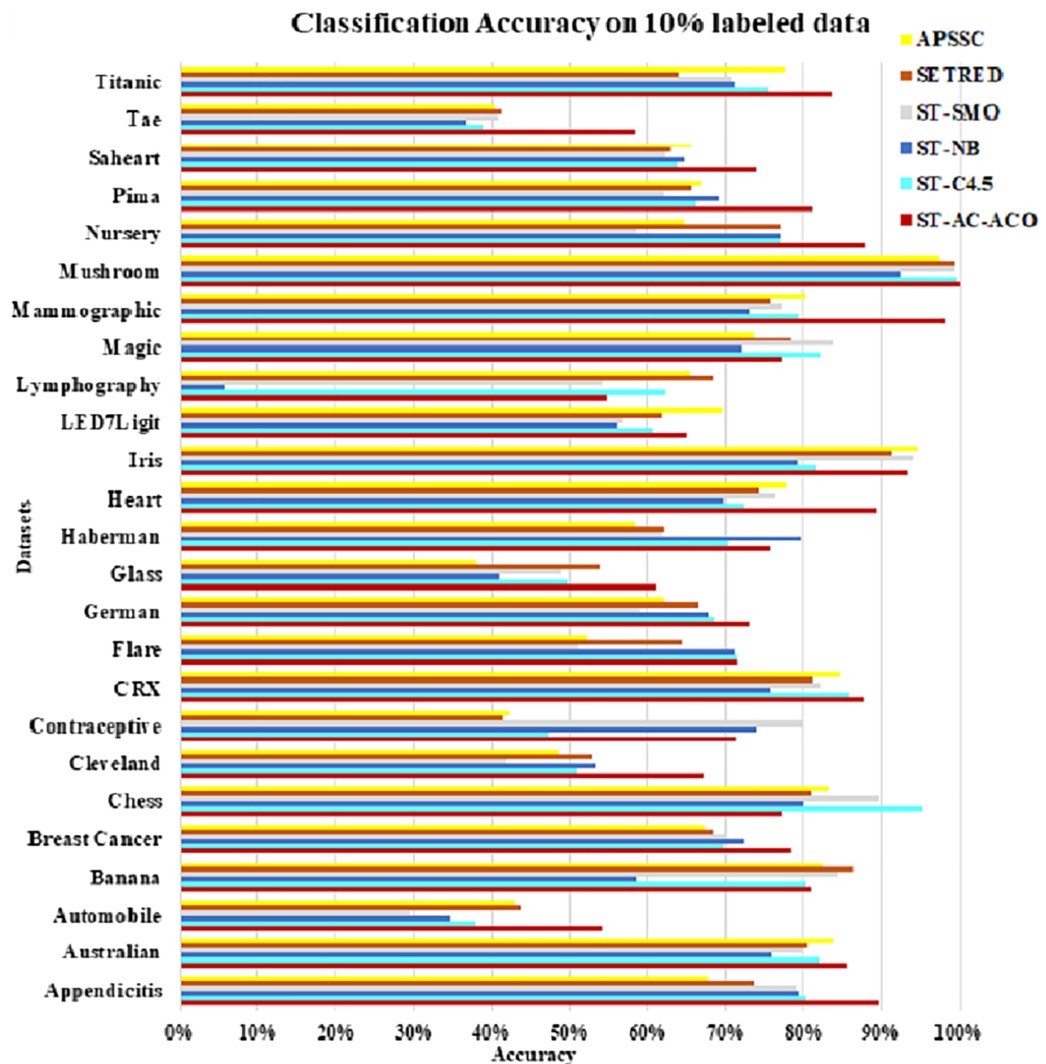

**Figure 6 Accuracy comparison of ST-AC-ACO with other self training algorithms (10% labeled data).**

*Lymphography* (with accuracy of 78.43%) datasets. ST-SMO suddenly dropped its lead that it attained against ST-AC-ACO on 20% labeled *Counterceptive* data and showed only 47.89%. This shows the lack of robustness of ST-SMO as compared to probabilistic approaches like ST-AC-ACO and ST-NB approaches. Wilcoxon tests show that ST-C4.5 and ST-SMO gave a little tougher competition to ST-AC-ACO despite the proposed approach still managed to show comprehensively higher accuracy on all of its competitors on 12 out of 25 datasets.

Table 10 accompanied by Figure 9 presents a comparative analysis of accuracies of self training algorithms on 40% labeled data. The proposed ST-AC-ACO algorithm attained 100% accuracy on *Chess* dataset beating all competitors. Similarly, ST-AC-ACO attained lead over all competitors on *Iris* dataset with an accuracy of 97.33%. Results on rest of the datasets remained almost unchanged. Wilcoxon test results show that

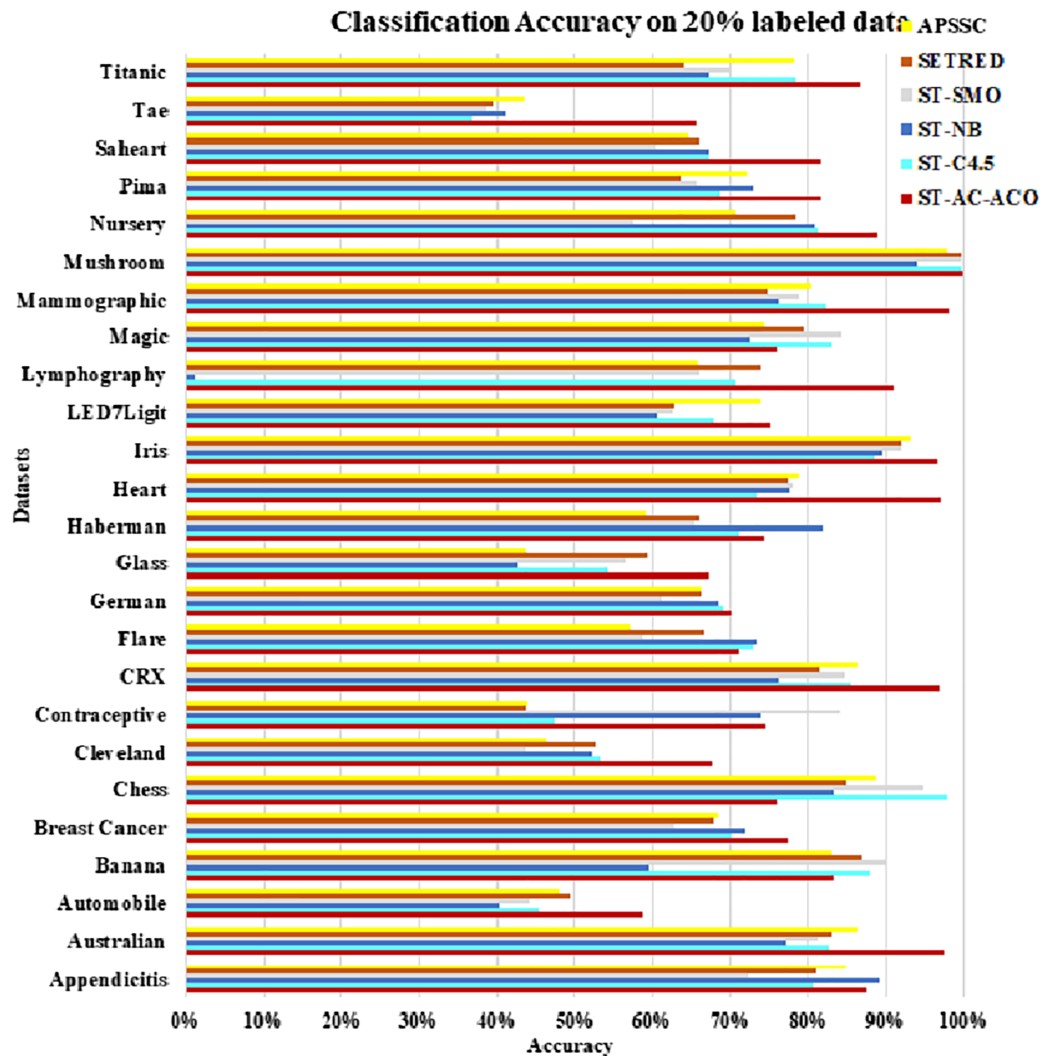

**Figure 7 Accuracy comparison of ST-AC-ACO with other self training algorithms (20% labeled data).**

ST-AC-ACO beat all non-associative competing classification algorithms comprehensively on 12 datasets.

To validate the results of experiments we performed statistical analysis using Wilcoxon Signed Rank Test (*Garca et al., 2010*). The reason to use this test instead of other statistical significance tests like pair-wise t-test is that it is non-parametric and makes no assumption about normal distribution of the data being analyzed. In our experimentation testing, the null hypothesis ($H_0$) states that there is no *significant* difference between the medians of accuracies (10-X folds) of ST-AC-ACO and each of its competitors on a specific dataset. The alternate hypothesis ($H_1$) states that there is a significant difference between medians of accuracies of ST-AC-ACO and each of its competitors on a specific dataset. When $H_0$ is not rejected, the accuracy of ST-AC-ACO is comparable (*Comp*) to that of its competitor. If $H_0$ is rejected and average accuracy of ST-AC-ACO is higher than that of its competitor, we conclude that accuracy of ST-AC-ACO is
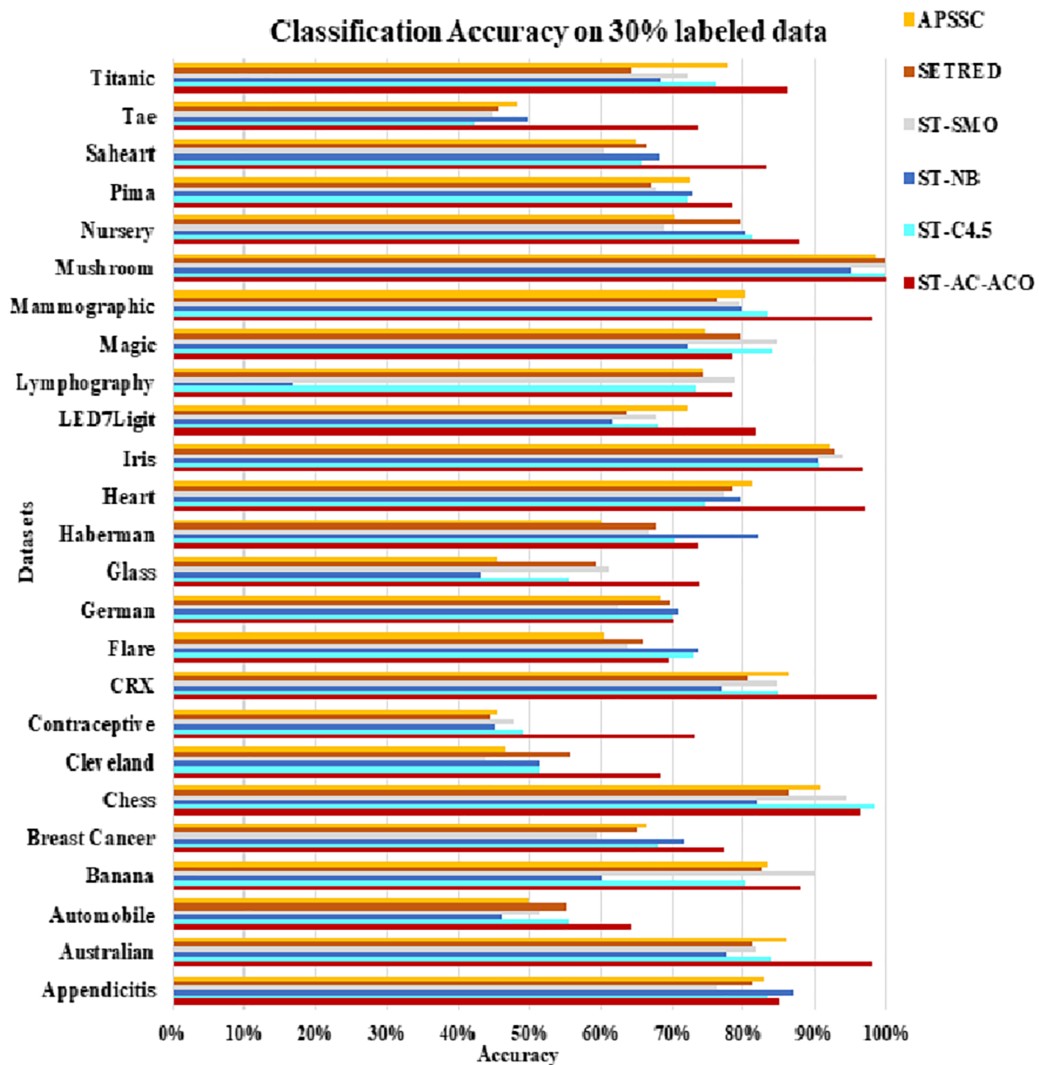

**Figure 8 Accuracy comparison of ST-AC-ACO with other self training algorithms (30% labeled data).**

significantly higher (*Win*), otherwise if the accuracy of ST-AC-ACO is lower than ots competitor while $H_0$ is rejected, we conclude that ST-AC-ACO showed significantly lower accuracy (*Loss*). The threshold ($w_{critical}$) is 8 for 10 readings (for 10-X fold validation). More details of the statistical test can be downloaded from the website (http://www. hamidawan.com.pk/research/).

Table 11 presents the significance analysis of comparison of ST-AC-ACO with its competitors on 10% labeled data. The bottom three lines describe the summary of wins, defeats and draws (*Comp*) achieved by ST-AC-ACO against each of its competitors. Positive *W* statistiv value in a comparison shows that score of ST-AC-ACO in the test was lower than that of its appropriate competitor while negative *W* indicates that the appropriate competitor has a lower *W* statistic value despite the insignificance of the difference. It is important to note that the proposed ST-AC-ACO beat all the competitors

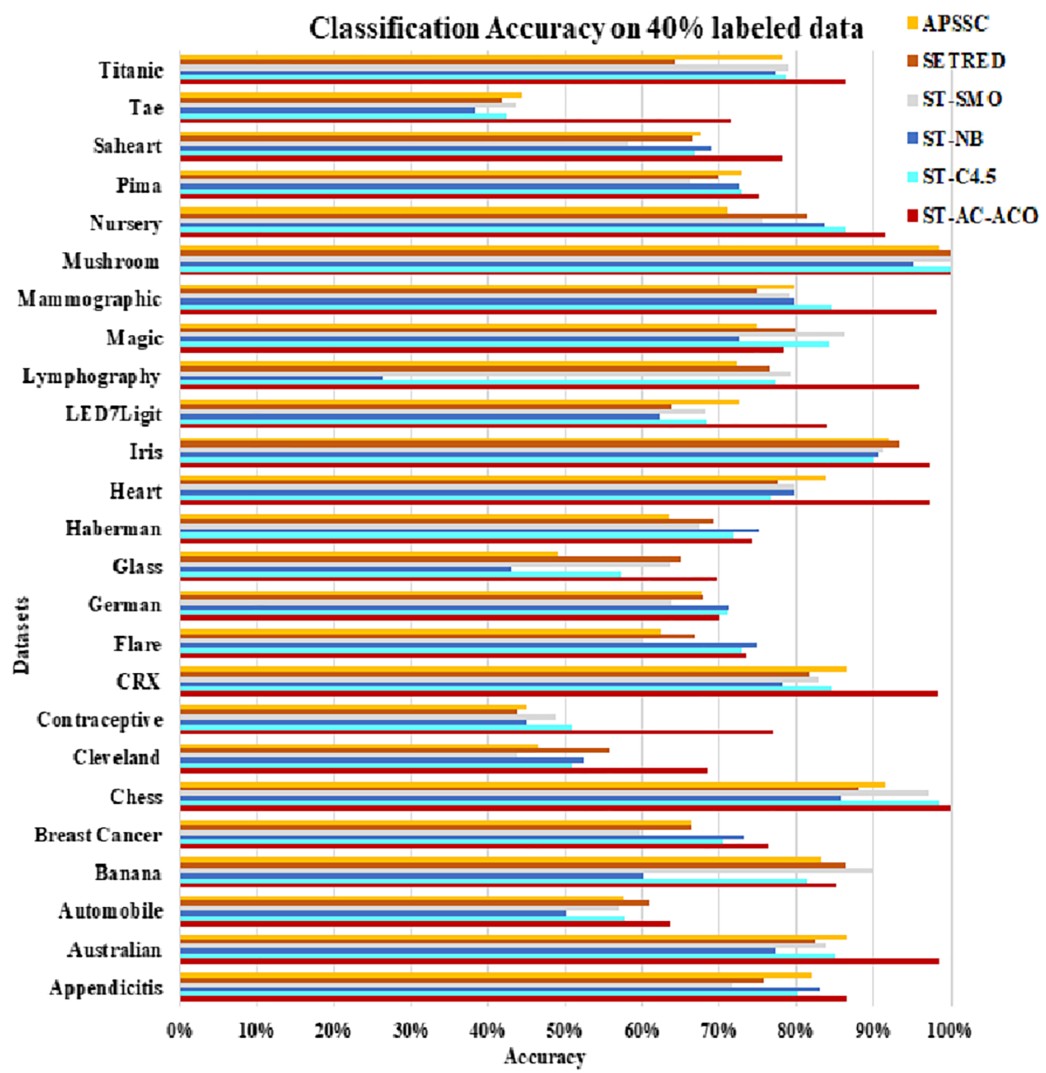

**Figure 9 Accuracy comparison of ST-AC-ACO with other self training algorithms (40% labeled data).**

on 10 datasets. ST-AC-ACO achieved a minimum of 14 wins (from SETRED) and faced a maximum of three defeats (from ST-C4.5).

Table 12 demonstrates the significance analysis of comparison on 20% labeled data. Negative *W* statistic value for a *Comp* result indicates that the appropriate competitor's score is less than that of ST-AC-ACO despite of the insignificance of the difference. ST-AC-ACO faces a maximum of four losses from ST-SMO and achieved a minimum of 14 wins from APSSC. The proposed ST-AC-ACO outperform all competitors with a significant difference in accuracy on 9 datasets.

Table 13 demonstrates the significance analysis of comparison on 30% labeled data. Negative *W* statistic value for a *Comp* result demonstrates that the appropriate competitor's score is less than that of ST-AC-ACO despite of the insignificance of the difference. ST-AC-ACO faced a maximum of two losses from ST-SMO and achieved a

**Table 8 Classification comparison on 20% labeled data.**

| Datasets | ST-AC-ACO (%) | ST-C4.5 (%) | ST-NB (%) | ST-SMO (%) | SETRED (%) | APSSC (%) |
|---|---|---|---|---|---|---|
| Appendicitis | 87.64 | 80.74 | 89.00 | 72.25 | 81.00 | 84.82 |
| Australian | 97.54 | 82.52 | 77.02 | 81.27 | 83.04 | 86.38 |
| Automobile | 58.63 | 45.34 | 40.23 | 44.26 | 49.51 | 47.90 |
| Banana | 83.30 | 88.04 | 59.47 | 89.85 | 86.91 | 82.91 |
| Breast Cancer | 77.30 | 70.22 | 71.97 | 62.95 | 67.88 | 68.53 |
| Chess | 76.13 | 97.78 | 83.26 | 94.84 | 84.89 | 88.55 |
| Cleveland | 67.69 | 53.11 | 52.18 | 43.72 | 52.57 | 46.24 |
| Contraceptive | 74.54 | 47.39 | 73.96 | 84.06 | 43.79 | 43.92 |
| CRX | 96.94 | 85.51 | 76.32 | 84.57 | 81.42 | 86.28 |
| Flare | 71.11 | 72.83 | 73.26 | 58.65 | 66.60 | 57.12 |
| German | 70.20 | 69.18 | 68.54 | 61.14 | 66.20 | 66.20 |
| Glass | 67.32 | 54.28 | 42.72 | 56.57 | 59.35 | 43.79 |
| Haberman | 74.49 | 70.96 | 81.92 | 65.43 | 66.00 | 59.08 |
| Heart | 97.04 | 73.44 | 77.54 | 77.85 | 77.41 | 78.89 |
| Iris | 96.67 | 88.43 | 89.44 | 91.94 | 92.00 | 93.33 |
| LED7Ligit | 75.20 | 67.94 | 60.64 | 62.72 | 62.80 | 74.00 |
| Lymphography | 91.05 | 70.65 | 1.23 | 66.02 | 73.92 | 65.69 |
| Magic | 76.11 | 73.04 | 64.49 | 74.21 | 69.34 | 64.30 |
| Mammographic | 98.07 | 82.43 | 76.33 | 78.87 | 74.73 | 80.46 |
| Mushroom | 100.00 | 99.83 | 94.11 | 99.77 | 99.84 | 97.87 |
| Nursery | 88.76 | 81.35 | 80.93 | 57.55 | 78.27 | 70.59 |
| Pima | 81.76 | 68.78 | 72.94 | 65.56 | 63.69 | 72.14 |
| Saheart | 81.81 | 67.16 | 67.22 | 60.39 | 66.01 | 64.50 |
| Tae | 65.54 | 36.82 | 41.06 | 38.74 | 39.63 | 43.58 |
| Titanic | 86.82 | 78.34 | 67.32 | 69.90 | 64.07 | 78.06 |

minimum of 16 wins from ST-C4.5. Moreover, ST-AC-ACO outperform all competitors with a significant difference in accuracy on 12 datasets.

Table 14 demonstrates the significance analysis of comparison on 40% labeled data. Negative *W* value for a *Comp* result indicates that ST-AC-ACO showed better performance than the appropriate competitor despite of the insignificance of the difference. ST-AC-ACO faced a maximum of two losses from ST-SMO and achieved a minimum of 17 wins from SETRED. The proposed ST-AC-ACO outperform all competitors with a significant difference in accuracy on 13 datasets.

The *Cohen's Kappa* measures (K statistics) were calculated to further evaluate the performance of the performance of ST-AC-ACO and its competitors. This measure is useful to validate the performance of classifiers on imbalanced datasets because accuracy may be misleading on such datasets. Each class is considered as the *rater* of the values of the confusion matrix of each classifier. K statistic attempts to reduce the portion of a

**Table 9 Classification comparison on 30% labeled data.**

| Datasets | ST-AC-ACO (%) | ST-C4.5 (%) | ST-NB (%) | ST-SMO (%) | SETRED (%) | APSSC (%) |
|---|---|---|---|---|---|---|
| Appendicitis | 85.09 | 83.38 | 87.08 | 76.27 | 81.09 | 82.91 |
| Australian | 98.12 | 83.82 | 77.59 | 81.56 | 81.01 | 85.94 |
| Automobile | 64.17 | 55.31 | 46.15 | 51.29 | 55.15 | 49.76 |
| Banana | 87.83 | 80.20 | 60.15 | 90.06 | 82.34 | 83.26 |
| Breast Cancer | 77.25 | 67.96 | 71.74 | 59.56 | 64.95 | 66.28 |
| Chess | 96.37 | 98.16 | 81.75 | 94.23 | 86.48 | 90.83 |
| Cleveland | 68.33 | 51.44 | 51.50 | 43.65 | 55.66 | 46.62 |
| Contraceptive | 73.12 | 48.95 | 45.06 | 47.89 | 44.26 | 45.42 |
| CRX | 98.77 | 84.82 | 76.93 | 84.70 | 80.67 | 86.31 |
| Flare | 69.34 | 72.89 | 73.56 | 63.79 | 65.94 | 60.50 |
| German | 70.00 | 69.86 | 70.79 | 62.37 | 69.60 | 68.40 |
| Glass | 73.85 | 55.43 | 43.09 | 61.05 | 59.33 | 45.45 |
| Haberman | 73.53 | 70.32 | 82.08 | 66.75 | 67.61 | 60.14 |
| Heart | 97.04 | 74.44 | 79.59 | 77.08 | 78.52 | 81.11 |
| Iris | 96.67 | 90.63 | 90.32 | 93.89 | 92.67 | 92.00 |
| LED7Ligit | 81.60 | 67.94 | 61.59 | 67.59 | 63.60 | 72.00 |
| Lymphography | 78.43 | 73.37 | 16.63 | 78.77 | 74.39 | 74.38 |
| Magic | 78.52 | 73.90 | 64.13 | 74.53 | 69.50 | 64.50 |
| Mammographic | 98.07 | 83.26 | 79.70 | 79.45 | 76.20 | 80.33 |
| Mushroom | 100.00 | 99.90 | 94.89 | 99.90 | 99.91 | 98.39 |
| Nursery | 87.69 | 81.11 | 80.15 | 68.71 | 79.57 | 70.43 |
| Pima | 78.52 | 72.00 | 72.78 | 67.68 | 66.94 | 72.53 |
| Saheart | 83.12 | 65.63 | 68.03 | 60.31 | 66.44 | 64.72 |
| Tae | 73.54 | 42.13 | 49.65 | 44.73 | 45.62 | 48.33 |
| Titanic | 86.14 | 76.13 | 68.35 | 72.10 | 64.07 | 77.74 |

classifier's accuracy attained by chance. The equation to find the K statistic is given by the Eq. (19):

$$K = \frac{P_0 - P_c}{1 - P_c} \tag{19}$$

where $P_0$ represents the actual accuracy of the classifier and $P_c$ denotes the accuracy by chance. To further explain this, let us consider the confusion matrix of ST-AC-ACO on 10% labeled *Titanic* dataset:

$$\begin{bmatrix} 352 & 359 \\ 0 & 1490 \end{bmatrix}$$

There are two classes (1 and −1). $P_0$ (accuracy) is calculated as sum of diagonal entries divided by number of all entries in the confusion matrix. Therefore

**Table 10 Classification comparison on 40% labeled data.**

| Datasets | ST-AC-ACO (%) | ST-C4.5 (%) | ST-NB (%) | ST-SMO (%) | SETRED (%) | APSSC (%) |
|---|---|---|---|---|---|---|
| Appendicitis | 86.64 | 80.18 | 82.91 | 71.55 | 75.64 | 82.00 |
| Australian | 98.55 | 85.07 | 77.39 | 83.77 | 82.46 | 86.52 |
| Automobile | 63.58 | 57.60 | 50.09 | 57.01 | 60.89 | 57.53 |
| Banana | 85.15 | 81.28 | 60.06 | 89.98 | 86.49 | 83.11 |
| Breast Cancer | 76.26 | 70.46 | 73.32 | 59.75 | 66.32 | 66.36 |
| Chess | 100.00 | 98.53 | 85.73 | 97.15 | 87.95 | 91.52 |
| Cleveland | 68.67 | 50.93 | 52.34 | 43.80 | 55.86 | 46.48 |
| Contraceptive | 76.92 | 50.85 | 45.01 | 48.68 | 43.79 | 45.08 |
| CRX | 98.47 | 84.65 | 78.18 | 82.82 | 81.61 | 86.64 |
| Flare | 73.46 | 72.80 | 74.77 | 59.85 | 66.79 | 62.47 |
| German | 70.10 | 71.00 | 71.30 | 63.70 | 67.90 | 67.80 |
| Glass | 69.57 | 57.22 | 43.10 | 63.46 | 64.87 | 49.22 |
| Haberman | 74.17 | 71.86 | 75.12 | 67.31 | 69.24 | 63.39 |
| Heart | 97.41 | 76.67 | 79.63 | 79.63 | 77.41 | 83.70 |
| Iris | 97.33 | 90.00 | 90.67 | 91.33 | 93.33 | 92.00 |
| LED7Ligit | 84.00 | 68.40 | 62.20 | 68.20 | 63.80 | 72.60 |
| Lymphography | 95.95 | 77.25 | 26.52 | 79.25 | 76.39 | 72.24 |
| Magic | 78.32 | 74.16 | 64.56 | 76.28 | 69.96 | 64.81 |
| Mammographic | 98.07 | 84.52 | 79.65 | 79.17 | 74.86 | 79.75 |
| Mushroom | 100.00 | 100.00 | 95.04 | 99.91 | 99.98 | 98.56 |
| Nursery | 91.51 | 86.42 | 83.64 | 75.49 | 81.41 | 71.02 |
| Pima | 75.12 | 72.92 | 72.55 | 66.25 | 69.80 | 72.80 |
| Saheart | 78.12 | 66.66 | 69.07 | 57.98 | 66.64 | 67.56 |
| Tae | 71.50 | 42.37 | 38.42 | 43.71 | 41.75 | 44.42 |
| Titanic | 86.37 | 78.74 | 77.33 | 78.83 | 64.07 | 78.06 |

$$P_0 = \frac{352 + 1490}{352 + 359 + 0 + 1490} = 0.8387 \tag{20}$$

Probability of chance (accuracy by chance) is calculated as:

$$P_c = \frac{\sum_{i=1}^{n} C_i \times R_i}{S^n} \tag{21}$$

where $n$ represents number of classes, $C_i$ represents sum of the elements of the $i$th column, $R_i$ represents the sum of elements of the $i$th row and $S$ represents the sum of all elements of the confusion matrix (size of the test data. For the example mentioned above:

$$P_c = \frac{352 \times 711 + 1849 \times 1490}{2201^2} = 0.6197 \tag{22}$$

Hence, K value can be calculated according to the Eq. (19). So

**Table 11 Wilcoxon signed rank test result on 10% labeled data-ACO *vs.* others ($w_{critical}$ = 8).**

| Dataset | vs ST-C4.5 | | vs ST-NB | | vs ST-SMO | | vs SETRRED | | vs APSSC | |
|---|---|---|---|---|---|---|---|---|---|---|
| | W | Result | W | Result | W | Result | W | Result | W | Result |
| Appendicitis | 9 | Comp | 6 | Win | 9 | Comp | 6 | Win | 3 | Win |
| Australian | 0 | Win | 0 | Win | 0 | Win | −18 | Comp | −23 | Comp |
| Automobile | 2 | Win | 3 | Win | 0 | Win | −12 | Comp | −11 | Comp |
| Banana | −24 | Comp | 0 | Win | 21 | Comp | 13 | Comp | 24 | Comp |
| Breast-cancer | 1 | Win | 3 | Win | 2 | Win | −11 | Comp | 4 | Win |
| Chess | 5 | Loss | −24 | Comp | 11 | Comp | 21 | Comp | 16 | Comp |
| Cleveland | 0 | Win | 0 | Win | 0 | Win | 4 | Win | 1 | Win |
| Contraceptive | 0 | Win | 20 | Comp | −12 | Comp | 0 | Win | 0 | Win |
| Crx | 0 | Win | 0 | Win | 0 | Win | −18 | Comp | 27 | Comp |
| Flare | −26 | Comp | 27 | Comp | 0 | Win | 7 | Win | 0 | Win |
| German | 0 | Win | 1 | Win | 0 | Win | 1 | Win | 0 | Win |
| Glass | 0 | Win | 0 | Win | 0 | Win | −14 | Comp | 1 | Win |
| Haberman | 7 | Win | −12 | Comp | 0 | Win | 0 | Win | 2 | Win |
| Heart | 0 | Win | 0 | Win | 0 | Win | 1 | Win | 0 | Win |
| Iris | 0 | Win | 1 | Win | 12 | Comp | −16 | Comp | 26 | Comp |
| LED7Ligi | 2 | Win | 1 | Win | 1 | Win | −23 | Comp | 16 | Comp |
| Lymphography | 1 | Loss | 0 | Win | 0 | Win | 3 | Loss | 10 | Comp |
| Magic | 2 | Loss | 0 | Win | 0 | Loss | 22 | Comp | −9 | Comp |
| Mammographic | 0 | Win | 0 | Win | 0 | Win | 0 | Win | 0 | Win |
| Mushroom | 0 | Win | 0 | Win | 0 | Win | 0 | Win | 0 | Win |
| Nursery | 0 | Win | 0 | Win | 0 | Win | 0 | Win | 0 | Win |
| Pima | 0 | Win | 0 | Win | 0 | Win | 1 | Win | 0 | Win |
| Saheart. | 1 | Win | 1 | Win | 1 | Win | 6 | Win | 6 | Win |
| Tae | 0 | Win | 0 | Win | 1 | Win | 4 | Win | 4 | Win |
| Titanic | 0 | Win | 0 | Win | 1 | Win | 3 | Win | 3 | Win |
| **Win** | **19** | | **21** | | **19** | | **14** | | **16** | |
| **Loss** | **3** | | **0** | | **1** | | **1** | | **0** | |
| **Comp** | **3** | | **4** | | **5** | | **10** | | **9** | |

$$K = \frac{0.8369 - 0.6204}{1 - 0.6204} = 0.5703 \tag{23}$$

Eq. (23) determines thai after removing the probability of accuracy by chance, the actual accuracy is approximately 0.57 (57%). Kappa statistic the accuracy that can be attributed to the classifier itself (*Ben-David, 2007*). In other words, K value determines the real performance of the classifier without merely relying merely on advantage of some baseness like frequency of a majority class in an imbalanced dataset. The K value 1 represents a perfect *agreement* between the *observed accuracy* and *expected accuracy*. Observed accuracy is the ratio of number of correctly classified instances to the total number of instances. The expected accuracy involves the ratio of class-wise accuracies to the total

**Table 12 Wilcoxon signed rank test result on 20% labeled data-ACO *vs.* others ($w_{critical}$ = 8).**

| Dataset | vs ST-C4.5 | | vs ST-NB | | vs ST-SMO | | vs SETRRED | | vs APSSC | |
|---|---|---|---|---|---|---|---|---|---|---|
| | W | Result | W | Result | W | Result | W | Result | W | Result |
| Appendicitis | 11 | Comp | −25 | Comp | 3 | Win | 14 | Comp | −22 | Comp |
| Australian | 0 | Win | 0 | Win | 0 | Win | 0 | Win | 0 | Win |
| Automobile | 7 | Win | 3 | Win | 8 | Comp | 18 | Comp | −10 | Comp |
| Banana | 1 | Loss | 0 | Win | 1 | Loss | 16 | Comp | −22 | Comp |
| Breast-cancer | 3 | Win | 4 | Win | 1 | Win | 3 | Win | −22 | Comp |
| Chess | 3 | Loss | 11 | Comp | 3 | Loss | 10 | Comp | 5 | Loss |
| Cleveland | 0 | Win | 0 | Win | 0 | Win | 0 | Win | −8 | Comp |
| Contraceptive | 0 | Win | 24 | Comp | 0 | Loss | 0 | Win | 5 | Loss |
| Crx | 0 | Win | 0 | Win | 0 | Win | 0 | Win | 0 | Win |
| Flare | −21 | Comp | −18 | Comp | 0 | Win | 9 | Comp | 0 | Win |
| German | 11 | Comp | 0 | Win | 0 | Win | 6 | Win | 0 | Win |
| Glass | 1 | Win | 0 | Win | 7 | Win | 3 | Win | 0 | Win |
| Haberman | 4 | Win | −2 | Loss | 0 | Win | 4 | Win | 5 | Win |
| Heart | 0 | Win | 0 | Win | 0 | Win | 0 | Win | 0 | Win |
| Iris | 8 | Comp | 3 | Win | 10 | Comp | 12 | Comp | 3 | Win |
| LED7Ligi | 11 | Comp | 4 | Win | 4 | Win | 0 | Win | 0 | Win |
| Lymphography | 0 | Win | 0 | Win | 0 | Win | 4 | Win | −13 | Comp |
| Magic | 2 | Loss | 0 | Win | 0 | Loss | 9 | Comp | −17 | Comp |
| Mammographic | 0 | Win | 0 | Win | 0 | Win | 0 | Win | −14 | Comp |
| Mushroom | 0 | Win | 0 | Win | 0 | Win | 0 | Win | 1 | Win |
| Nursery | 0 | Win | 0 | Win | 0 | Win | 0 | Win | 0 | Win |
| Pima | 2 | Win | 3 | Win | 0 | Win | 1 | Win | −17 | Comp |
| Saheart. | 0 | Win | 0 | Win | 0 | Win | 1 | Win | 0 | Win |
| Tae | 0 | Win | 2 | Win | 1 | Win | 0 | Win | 0 | Win |
| Titanic | 0 | Win | 0 | Win | 0 | Win | 0 | Win | 0 | Win |
| **Win** | **17** | | **20** | | **19** | | **18** | | **14** | |
| **Loss** | **3** | | **1** | | **4** | | **0** | | **2** | |
| **Comp** | **5** | | **4** | | **2** | | **7** | | **9** | |

number of instances. Thus If a higher observed accuracy is merely due to the higher frequency of a majority class, the expected accuracy and consequently, the K value will be close to or even equal to 0, indicating a strong disagreement.

Table 15 demonstrates the comparison of ST-AC-ACO with respect to K values. The K value of ST-AC-ACO of 1.00 on *Mushroom* dataset shows a strong agreement between the expected and the observed accuracies as all classes were correctly classified. Similarly, the K-values of 0.00 for ST-AC-ACO on *Automobile* and *Lymphography* demonstrate that the classifier's accuracy was merely due to the frequency of the majority class. ST-AC-ACO beat all competitors on ten datasets comprehensively (see Table 11) on 10% labeled data but the K values suggest that three of such wins were by chance (*Loss* of 3). These datasets include *German*, *Nursery* and *Prima* because unlike the accuracy shown by

**Table 13 Wilcoxon signed rank test result on 30% labeled data-ACO *vs.* others ($w_{critical}$ = 8).**

| Dataset | vs ST-C4.5 | | vs ST-NB | | vs ST-SMO | | vs SETRRED | | vs APSSC | |
|---|---|---|---|---|---|---|---|---|---|---|
| | W | Result | W | Result | W | Result | W | Result | W | Result |
| Appendicitis | 15 | Comp | 24 | Comp | 6 | Win | −16 | Comp | −20 | Comp |
| Australian | 0 | Win | 0 | Win | 0 | Win | 0 | Win | 0 | Win |
| Automobile | 13 | Comp | 8 | Comp | 16 | Comp | −10 | Comp | −9 | Comp |
| Banana | 0 | Win | 0 | Win | 3 | Loss | 1 | Win | 0 | Win |
| Breast-cancer | 1 | Win | 2 | Win | 1 | Win | 1 | Win | 2 | Win |
| Chess | −19 | Comp | 3 | Win | −10 | Comp | −10 | Comp | −10 | Comp |
| Cleveland | 0 | Win | 0 | Win | 0 | Win | 5 | Win | 0 | Win |
| Contraceptive | 0 | Win | 0 | Win | 0 | Win | 0 | Win | 0 | Win |
| Crx | 0 | Win | 0 | Win | 0 | Win | 0 | Comp | 0 | Win |
| Flare | −20 | Comp | −17 | Comp | 1 | Win | −16 | Win | 7 | Win |
| German | 23 | Comp | −12 | Comp | 0 | Win | −25 | Win | −18 | Comp |
| Glass | 1 | Win | 0 | Win | 16 | Comp | 3 | Comp | 0 | Win |
| Haberman | 8 | Comp | −2 | Loss | 0 | Win | −10 | Win | 0 | Win |
| Heart | 0 | Win | 0 | Win | 0 | Win | 0 | Win | 0 | Win |
| Iris | 11 | Comp | 2 | Win | 10 | Comp | 7 | Win | 5 | Win |
| LED7Ligi | 10 | Comp | 5 | Win | 10 | Comp | 0 | Win | 0 | Win |
| Lymphography | 0 | Win | 0 | Win | 3 | Win | 1 | Win | 0 | Win |
| Magic | 0 | Loss | 0 | Win | 0 | Loss | 19 | Comp | 3 | Win |
| Mammographic | 0 | Win | 0 | Win | 0 | Win | 0 | Win | 0 | Win |
| Mushroom | 0 | Win | 0 | Win | 0 | Win | 0 | Win | 0 | Win |
| Nursery | 0 | Win | 0 | Win | 0 | Win | 0 | Win | 0 | Win |
| Pima | 2 | Win | 3 | Win | 0 | Win | 0 | Win | 4 | Win |
| Saheart. | 0 | Win | 0 | Win | 0 | Win | 1 | Win | 2 | Win |
| Tae | 1 | Win | 8 | Comp | 1 | Win | 1 | Win | 0 | Win |
| Titanic | 0 | Win | 0 | Win | 0 | Win | 4 | Win | 1 | Win |
| **Win** | **16** | | **19** | | **18** | | **19** | | **21** | |
| **Loss** | **1** | | **1** | | **2** | | **0** | | **0** | |
| **Comp** | **8** | | **5** | | **5** | | **6** | | **4** | |

ST-AC-ACO on these data sets (see Table 7), K values for the proposed classifier is not the *highest* on each of these datasets. However, on four datasets (*Appendicitis*, *Australian*, *Contraceptive* and *Magic*), ST-AC-ACO achieved the *highest* K values despite not being able to comprehensively beat all competitors with respect to observed accuracy (*Gain* of 4). So effectively, the proposed ST-AC-ACO performed better than all of its competitors on 11 datasets with respect to K value.

Table 16 demonstrates the comparison of K values of SY = AC-ACO and its competitors on 20% labeled datasets. ST-AC-ACO lost its lead in K values on three datasets (*CRX*, *Class* and *Saheart*) for which it had a significant lead in observed accuracy (see Tables 8 and 12). More notably, ST-AC-ACO achieved the highest K values on nine datasets for which its observed accuracy wasn't the highest on 20% labeled data.

**Table 14 Wilcoxon Signed Rank Test Result on 40% labeled data-ACO *vs.* others ($w_{critical}$ = 8).**

| Dataset | vs ST-C4.5 | | vs ST-NB | | vs ST-SMO | | vs SETRRED | | vs APSSC | |
|---|---|---|---|---|---|---|---|---|---|---|
| | W | Result | W | Result | W | Result | W | Result | W | Result |
| Appendicitis | 0 | Win | 1 | Win | 0 | Win | 7 | Win | −18 | Comp |
| Australian | 0 | Win | 0 | Win | 0 | Win | 0 | Win | 0 | Win |
| Automobile | −22 | Comp | 24 | Comp | −23 | Comp | −21 | Comp | −17 | Comp |
| Banana | −14 | Comp | 0 | Win | 5 | Loss | −19 | Comp | −19 | Comp |
| Breast-cancer | 0 | Win | 0 | Win | 0 | Win | 6 | Win | 7 | Win |
| Chess | 0 | Win | 0 | Win | 0 | Win | 0 | Win | 0 | Win |
| Cleveland | 0 | Win | 0 | Win | 0 | Win | 1 | Win | 0 | Win |
| Contraceptive | 0 | Win | 0 | Win | 0 | Win | 0 | Win | 0 | Win |
| Crx | 0 | Win | 0 | Win | 0 | Win | 0 | Win | 0 | Win |
| Flare | 7 | Win | 13 | Comp | 0 | Win | 2 | Win | 0 | Win |
| German | 0 | Win | 0 | Win | 0 | Win | −9 | Comp | −13 | Comp |
| Glass | 0 | Win | 0 | Win | 8 | Comp | −17 | Comp | 1 | Win |
| Haberman | −22 | Comp | 0 | Loss | 27 | Comp | −9 | Comp | 2 | Win |
| Heart | 0 | Win | 0 | Win | 1 | Win | 3 | Win | 0 | Win |
| Iris | 4 | Win | 1 | Win | 5 | Win | 5 | Win | 5 | Win |
| LED7Ligi | 11 | Comp | 6 | Win | 6 | Win | 0 | Win | 4 | Win |
| Lymphography | 0 | Win | 0 | Win | 0 | Win | 0 | Win | 0 | Win |
| Magic | 0 | Loss | 1 | Win | 0 | Loss | 15 | Comp | 7 | Win |
| Mammographic | 0 | Win | 0 | Win | 0 | Win | 0 | Win | 0 | Win |
| Mushroom | 0 | Comp | 0 | Win | 0 | Win | 0 | Win | 0 | Win |
| Nursery | 4 | Win | 0 | Win | 0 | Win | 0 | Win | 0 | Win |
| Pima | 0 | Win | 0 | Win | 0 | Win | 10 | Comp | −19 | Comp |
| Saheart. | 0 | Win | 0 | Win | 0 | Win | 9 | Comp | 2 | Win |
| Tae | 2 | Win | 2 | Win | 5 | Win | 1 | Win | 0 | Win |
| Titanic | 0 | Win | 0 | Win | 0 | Win | 0 | Win | 0 | Win |
| **Win** | **19** | | **22** | | **20** | | **17** | | **20** | |
| **Loss** | **1** | | **1** | | **2** | | **0** | | **0** | |
| **Comp** | **5** | | **2** | | **3** | | **8** | | **5** | |

Table 17 demonstrates the comparison of K values of ST-AC-ACO and its competitors on 30% labeled datasets. ST-AC-ACO lost in K values on the same three datasets as those on 20% labeled datasets. Moreover, ST-AC-ACO gained the highest K values on six datasets on which it didn't show the highest (significant) observed accuracy (see Tables 9 and 13). The difference of K values of ST-AC-ACO and its competitors in not just *marginal* on *Appendicitis, Australian, Cleveland, Contraceptive, Heart, Iris, Lymphography, Tae* and *Titanic*.

Table 18 demonstrates the comparison of K values of ST-AC-ACO and its competitors on 40% labeled datasets. ST-AC-ACO maintained its lead in K values against its competitors on all those 13 datasets on which it had shown a significant lead in accuracy

**Table 15 Analysis of Cohen's Kappa statistic on 10% labeled data.**

| Datasets | ST-AC-ACO | ST-C4.5 | ST-NB | ST-SMO | SETRED | APSSC |
|---|---|---|---|---|---|---|
| Appendicitis | **0.68** | 0.16 | 0.00 | 0.15 | 0.10 | 0.34 |
| Australian | **0.85** | 0.65 | 0.50 | 0.58 | 0.60 | 0.67 |
| Automobile | 0.00 | 0.21 | 0.11 | 0.03 | 0.25 | **0.26** |
| Banana | 0.60 | 0.69 | 0.14 | **0.79** | 0.72 | 0.65 |
| Breast Cancer | 0.29 | 0.15 | **0.30** | 0.00 | 0.15 | 0.19 |
| Chess | 0.30 | 0.91 | 0.60 | 0.79 | 0.62 | 0.66 |
| Cleveland | **0.47** | 0.25 | 0.19 | 0.20 | 0.27 | 0.27 |
| Contraceptive | **0.60** | 0.20 | **0.17** | 0.15 | 0.10 | 0.14 |
| CRX | **0.70** | **0.70** | 0.54 | 0.66 | 0.62 | 0.69 |
| Flare | 0.28 | **0.64** | 0.63 | 0.42 | 0.55 | 0.41 |
| German | 0.10 | 0.13 | 0.21 | **0.24** | 0.17 | 0.20 |
| Glass | 0.26 | 0.28 | 0.19 | 0.31 | **0.35** | 0.21 |
| Haberman | 0.11 | 0.11 | **0.27** | 0.09 | 0.08 | 0.10 |
| Heart | **0.85** | 0.35 | 0.40 | 0.55 | 0.48 | 0.56 |
| Iris | 0.89 | 0.76 | 0.67 | 0.91 | 0.87 | **0.92** |
| LED7Ligit | 0.19 | 0.57 | 0.50 | 0.52 | 0.57 | **0.66** |
| Lymphography | 0.00 | 0.38 | −0.01 | 0.00 | 0.38 | **0.43** |
| Magic | **0.42** | 0.39 | 0.30 | 0.39 | 0.34 | 0.34 |
| Mammographic | **0.92** | 0.61 | 0.47 | 0.56 | 0.52 | 0.61 |
| Mushroom | **1.00** | 0.99 | 0.84 | 0.98 | 0.99 | 0.95 |
| Nursery | **0.81** | 0.76 | 0.68 | 0.55 | 0.72 | 0.58 |
| Pima | 0.10 | 0.26 | 0.34 | 0.24 | 0.27 | **0.40** |
| Saheart | 0.15 | 0.24 | **0.32** | 0.21 | 0.19 | 0.28 |
| Tae | **0.11** | 0.10 | 0.10 | 0.10 | 0.08 | 0.10 |
| Titanic | **0.57** | 0.41 | 0.44 | 0.42 | 0.27 | 0.44 |

**Note:**
Entries in boldface in each row represent winning value(s).

(see Tables 10 and 14). Moreover, ST-AC-ACO showed highest K scores on four additional datasets on which it didn't show a significant observed classification accuracy.

## DISCUSSION

The proposed algorithm ST-AC-ACO showed promising results on the majority of datasets used in the experimentation. Tables 7–10 demonstrate the accuracy comparison of ST-AC-ACO with its competing techniques. Wilcoxon Signed Rank test results shown in Tables 11–14 demonstrate that ST-AC-ACO comprehensively outperformed its competitors with respect to accuracy.

Cohen's Kappa statistic was calculated as a statistical test on confusion matrices resulted from 10-fold cross-validation execution of ST-AC-ACO and its competitors. The results of the K statistics show that ST-AC-ACO still outperformed the majority of its competitors on most datasets.

**Table 16 Analysis of Cohen's Kappa statistic on 20% labeled data.**

| Datasets | ST-AC-ACO | ST-C4.5 | ST-NB | ST-SMO | SETRED | APSSC |
|---|---|---|---|---|---|---|
| Appendicitis | **0.72** | 0.33 | 0.55 | 0.18 | 0.42 | 0.57 |
| Australian | **0.94** | 0.71 | 0.53 | 0.66 | 0.66 | 0.73 |
| Automobile | **0.47** | 0.31 | 0.14 | 0.27 | 0.33 | 0.33 |
| Banana | 0.65 | 0.76 | 0.15 | **0.79** | 0.73 | 0.65 |
| Breast Cancer | **0.40** | 0.16 | 0.33 | 0.24 | 0.17 | 0.23 |
| Chess | 0.70 | **0.96** | 0.67 | 0.90 | 0.70 | 0.77 |
| Cleveland | **0.47** | 0.20 | 0.28 | 0.23 | 0.25 | 0.24 |
| Contraceptive | **0.62** | 0.19 | 0.19 | 0.18 | 0.14 | 0.18 |
| CRX | 0.44 | 0.71 | 0.55 | 0.72 | 0.62 | **0.73** |
| Flare | 0.33 | 0.65 | **0.66** | 0.49 | 0.58 | 0.47 |
| German | 0.00 | 0.18 | 0.27 | **0.28** | 0.18 | 0.26 |
| Glass | 0.29 | 0.33 | 0.24 | 0.40 | **0.44** | 0.26 |
| Haberman | 0.04 | 0.14 | **0.24** | 0.17 | 0.11 | 0.10 |
| Heart | **0.62** | 0.49 | 0.60 | 0.58 | 0.54 | 0.57 |
| Iris | **0.95** | 0.84 | 0.86 | 0.87 | 0.88 | 0.90 |
| LED7Ligit | 0.12 | 0.64 | 0.53 | 0.60 | 0.59 | **0.71** |
| Lymphography | **0.89** | 0.43 | 0.00 | 0.24 | 0.48 | 0.39 |
| Magic | **0.38** | 0.38 | 0.30 | 0.37 | 0.37 | 0.31 |
| Mammographic | **0.96** | 0.65 | 0.55 | 0.59 | 0.49 | 0.61 |
| Mushroom | **1.00** | 1.00 | 0.87 | 0.99 | 1.00 | 0.95 |
| Nursery | **0.76** | 0.69 | 0.69 | 0.47 | 0.65 | 0.62 |
| Pima | 0.04 | 0.31 | **0.41** | 0.30 | 0.20 | 0.39 |
| Saheart | 0.31 | 0.20 | **0.35** | 0.22 | 0.26 | 0.28 |
| Tae | **0.29** | 0.14 | 0.12 | 0.13 | 0.09 | 0.16 |
| Titanic | **0.63** | 0.42 | 0.44 | 0.42 | 0.27 | 0.45 |

**Note:**
Entries in boldface in each row represent winning value(s).

Performance of ST-AC-ACO on larger datasets like *Mushroom*, *Nursery* and *Titanic* has been significantly better than all of its competitors on all proportions of labeled datasets. On the other hand, ST-AS-ACO didn't perform better than all of its competitors on 10%, 20% and 30% labeled proportions *Chess* dataset. It showed an improved performance on *Banana*, *Chess* and *Magic* datasets when 30% or more labeled data was presented.

It has been shown that the power of *associative property* makes associative classification much more robust and reliable than other non-associative classifiers in self-trained semi-supervised classification. The discovery of frequent patterns allows classification be more accurate and robust than merely constructing classification rules without considering association among non-class attributes. The results showed by the ST-AC-ACO are according to the expectation that associative classification should perform better in semi-supervised learning as it did in supervised learning in prior proposed algorithms (*Hadi, Al-Radaideh & Alhawari, 2018*; *Shahzad & Baig, 2011*).

**Table 17 Analysis of Cohen's Kappa statistic on 30% labeled data.**

| Datasets | ST-AC-ACO | ST-C4.5 | ST-NB | ST-SMO | SETRED | APSSC |
|---|---|---|---|---|---|---|
| Appendicitis | **0.72** | 0.44 | 0.54 | 0.30 | 0.49 | 0.53 |
| Australian | **0.98** | 0.69 | 0.52 | 0.62 | 0.61 | 0.72 |
| Automobile | **0.43** | 0.41 | 0.29 | 0.34 | 0.41 | 0.36 |
| Banana | 0.60 | 0.76 | 0.16 | **0.80** | 0.74 | 0.66 |
| Breast Cancer | **0.37** | 0.16 | 0.31 | 0.23 | 0.13 | 0.22 |
| Chess | 0.95 | **0.97** | 0.71 | 0.93 | 0.73 | 0.82 |
| Cleveland | **0.46** | 0.26 | 0.28 | 0.25 | 0.29 | 0.24 |
| Contraceptive | **0.63** | 0.19 | 0.19 | 0.20 | 0.15 | 0.20 |
| CRX | **0.72** | 0.70 | 0.54 | 0.70 | 0.61 | 0.71 |
| Flare | 0.36 | 0.65 | **0.66** | 0.56 | 0.57 | 0.51 |
| German | 0.00 | 0.25 | 0.31 | 0.29 | 0.26 | **0.31** |
| Glass | 0.30 | 0.37 | 0.24 | 0.44 | **0.44** | 0.31 |
| Haberman | 0.04 | 0.14 | **0.22** | 0.15 | 0.13 | 0.10 |
| Heart | **0.91** | 0.50 | 0.64 | 0.62 | 0.56 | 0.62 |
| Iris | **0.96** | 0.88 | 0.86 | 0.90 | 0.89 | 0.88 |
| LED7Ligit | 0.12 | 0.65 | 0.58 | 0.65 | 0.59 | **0.69** |
| Lymphography | **0.72** | 0.54 | 0.00 | 0.59 | 0.51 | 0.53 |
| Magic | **0.39** | 0.38 | 0.32 | 0.36 | 0.34 | 0.33 |
| Mammographic | **0.96** | 0.69 | 0.60 | 0.59 | 0.52 | 0.61 |
| Mushroom | **1.00** | 1.00 | 0.88 | 1.00 | 1.00 | 0.97 |
| Nursery | 0.76 | **0.81** | 0.75 | 0.61 | 0.76 | 0.64 |
| Pima | 0.07 | **0.40** | 0.38 | 0.29 | 0.28 | 0.39 |
| Saheart | 0.33 | 0.25 | 0.36 | **0.18** | 0.27 | 0.28 |
| Tae | **0.35** | 0.14 | 0.22 | 0.25 | 0.19 | 0.23 |
| Titanic | **0.64** | 0.42 | 0.44 | 0.44 | 0.27 | 0.45 |

**Note:**
Entries in boldface in each row represent winning value(s).

Moreover, ST-AC-ACO uses its own mechanism for weighing pseudo-labeled instances in transductive learning which balances out the bias of frequent terms that mostly occur in pseudo-labeled instances. This reduces the overall impact of any incorrect pseudo-labeling.

The proposed approach has its application (like other SSL methods) in information retrieval (like web page classification, social media mining, etc), bio-informatics (such as protein classification), business strategy planning and robotics where only a small portion of data is labeled. The incorporation of associative classification is expected to increase the performance of classification in such applications.

ST-AC-ACO however, needs to he optimized for dealing with highly imbalanced data. This is one of the future research direction to extend the proposed ST-AC-ACO algorithm. Similarly, application of the proposed algorithm can be performed on high-dimensional real-life complex problems like the one addressed in *Fu et al. (2020)*. Moreover, empirical studies like impact of feature subset selection, feature extraction, etc can also be performed.

**Table 18 Analysis of Cohen's Kappa statistic on 40% labeled data.**

| Datasets | ST-AC-ACO | ST-C4.5 | ST-NB | ST-SMO | SETRED | APSSC |
|---|---|---|---|---|---|---|
| Appendicitis | **0.59** | 0.43 | 0.47 | 0.30 | 0.39 | 0.44 |
| Australian | **0.98** | 0.70 | 0.53 | 0.67 | 0.64 | 0.56 |
| Automobile | **0.47** | 0.45 | 0.35 | 0.42 | 0.45 | 0.46 |
| Banana | 0.71 | 0.76 | 0.16 | **0.80** | 0.73 | 0.66 |
| Breast Cancer | **0.36** | 0.13 | 0.31 | 0.22 | 0.17 | 0.23 |
| Chess | **1.00** | 0.97 | 0.71 | 0.94 | 0.76 | 0.83 |
| Cleveland | **0.48** | 0.23 | 0.29 | 0.23 | 0.30 | 0.23 |
| Contraceptive | **0.58** | 0.24 | 0.19 | 0.20 | 0.14 | 0.20 |
| CRX | **0.98** | 0.69 | 0.55 | 0.66 | 0.63 | 0.73 |
| Flare | 0.36 | 0.65 | **0.68** | 0.50 | 0.58 | 0.53 |
| German | 0.01 | 0.27 | 0.29 | **0.31** | 0.22 | 0.30 |
| Glass | 0.28 | 0.42 | 0.22 | 0.51 | **0.52** | 0.35 |
| Haberman | 0.00 | 0.16 | **0.26** | 0.14 | 0.16 | 0.16 |
| Heart | **0.89** | 0.53 | 0.59 | 0.59 | 0.54 | 0.67 |
| Iris | **0.96** | 0.85 | 0.86 | 0.87 | 0.90 | 0.88 |
| LED7Ligit | 0.17 | 0.65 | 0.58 | 0.65 | 0.60 | **0.69** |
| Lymphography | **0.93** | 0.57 | 0.00 | 0.59 | 0.52 | 0.52 |
| Magic | **0.47** | 0.43 | 0.33 | 0.39 | 0.35 | 0.33 |
| Mammographic | **0.96** | 0.69 | 0.59 | 0.58 | 0.50 | 0.60 |
| Mushroom | **1.00** | 1.00 | 0.89 | 1.00 | 1.00 | 0.97 |
| Nursery | **0.87** | 0.86 | 0.82 | 0.70 | 0.76 | 0.63 |
| Pima | 0.04 | 0.37 | **0.39** | 0.30 | 0.32 | 0.39 |
| Saheart | 0.32 | 0.24 | **0.35** | 0.19 | 0.26 | 0.33 |
| Tae | **0.39** | 0.13 | 0.08 | 0.16 | 0.13 | 0.17 |
| Titanic | **0.65** | 0.43 | 0.44 | 0.44 | 0.27 | 0.46 |

**Note:**
Entries in boldface in each row represent winning value(s).

# CONCLUSION

A novel rule-based semi-supervised associative classification approach using ant colony optimization has been proposed in this article. The primary task of the approach is to learn from a smaller ratio of labeled data than unlabeled data to first label the unlabeled data and then apply the classification rules. This approach uses labeled data to first discover associative classification rules with ACO and then using those rules in transductive mechanism to label the unlabeled instances. The experimental results of the proposed technique demonstrate that the proposed ST-AC-ACO algorithm is not only superior in accuracy to its competing self-training algorithms but it is more robust as it tends to discover relationship between a frequent itemset of non-class attributes and the class labels. This approach can further be combined with feature subset selection to remove unnecessary or redundant attributes for even better classification accuracy. Moreover, the proposed approach can also be utilized for labeling and classification of big data with a little fraction of labeled data. Another future direction is to develop a mechanism to find frequent patterns from the entire (labeled and unlabeled) dataset and assign the most

confident class. A more fundamental task in this regard is to re-define the SSL problem by omitting the classification and presenting results just based on pseudo-labeling, after all classification is a secondary task and its performance directly depends on pseudo-labeling of unlabeled instances.

### Funding
The authors received no funding for this work.

### Competing Interests
The authors declare that they have no competing interests.

### Author Contributions
- Hamid Hussain Awan conceived and designed the experiments, analyzed the data, performed the computation work, prepared figures and/or tables, authored or reviewed drafts of the paper, and approved the final draft.
- Waseem Shahzad performed the computation work, authored or reviewed drafts of the paper, supervision and validation of results, and approved the final draft.

### Data Availability
The code is available in the Supplemental File.

The implementation of ST-AC-ACO does not involve KEEL. The KEEL tool was used to run the competing algorithms for comparison. The reference manual for KEEL software is available at: https://sci2s.ugr.es/keel/development.php.

### Supplemental Information
Supplemental information for this article can be found online at http://dx.doi.org/10.7717/peerj-cs.676#supplemental-information.

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
