# Peer review of "Semi-supervised associative classification using ant colony optimization algorithm"

_PeerJ Computer Science, doi:10.7717/peerj-cs.676_

## Round 0.1 · original submission · Major Revisions

The two reviewers both have major concerns. As a result, a major revision is recommended.

Reviewer 1 ·

Basic reporting

no comment

Experimental design

no comment

Validity of the findings

no comment

Additional comments

The paper topic is timely, structure wise is nice, and writing style is understandable and readable. Performance evaluation is perfect and sufficient to prove the system behaviour. So, no concerns with the paper from these perspectives. However, the paper suffers some major issues as below:
1. I don’t seem to be able to figure out where the list of novelty is. Please make sure you have clearly identified them in the first section.
2. Please pay more attention to the motivation of the work.
3. The authors should evaluate and include computational complexity of the proposed approach using asymptotic measurements.
4. The authors should use statistical analysis to determine the significant difference among the competing methods. There are some statistical methods for analyzing competing classifiers over multiple data sets.
5. Train and test validation is utilized to compute the accuracy in the comparison. It has been proved that cross-validation is most adequate for comparing supervised classification algorithms than holdout validation (train and test) Cross-validation is used to overcome the limitations of the holdout validation , that is why it makes no sense to compute the accuracy using holdout validation. The correct method to compute the accuracy (or other measures) is cross-validation or repeated cross-validation.
6. The description is unclear on how to train the model. For example, the description of the training data set is lacking. Without this, it is hard to tell if the method is implemented properly.
7. The evaluation is not sufficient. The authors may provide an adequate evaluation by using a larger dataset and comparing it to more state-of-the-art methods. With which, the authors may improve the impact of the work as well.
8. Theoretical explanations are very less in proposed approach.
9. Critical analysis with comparison with state of the art would have raised the quality of the paper.
10. Proofread the paper and double check the information of the references for completeness.

Reviewer 2 ·

Basic reporting

If possible, could you please describe how would the Rule-Based Semi-Supervised Associative Classification Approach and Ant Colony Optimization (ACO) work. Could you update more details of the principles and fundamentals of Unsupervised Learning, Semi-Supervised Learning, and Supervised Learning algorithms?

Figure 1, Page 8:

Could you respectively update more descriptions in Figure 1 on Page 8 from T1 to T6, and the “Infinite Component” at the bottom? What are the representations about Solid Line and Dotted Line? If possible, could you please mark these information mentioned above into the Figure 1?

Experimental design

The details of training data and testing data should be listed and respectively demonstrated the percentage of training data and testing data of the original dataset in terms of case studies or/and experimentations.

Line 195, Page 9:

Authors mentioned “Algorithm 1 illustrates the proposed ST-AC-ACO algorithm.” in Line 195 on Page 9. However “Algorithm 1 SSAC-ACO” is shown on Page 11, which cannot be efficaciously followed by readers.

The proposed algorithm ST-AC-ACO should be demonstrated as the form of Algorithm Environment, which can be efficaciously followed by readers. Additionally, the flowchart of proposed method ST-AC-ACO mentioned in this paper should also be updated and included the dataset collections, pre-processing of dataset, post-processing of dataset, feature extractions, feature selections, and the performances of fault classification.

Alternatively, different topologies of experimental samples and the number of significant feature selections might have either positive or negative impacts for the performances of fault classification for more complicated industrial systems to some extent. Based on this reason, could you please provide an evidence to make explanations the availability and feasibility of the proposed methodology based on ST-AC-ACO in the field of Semi-Supervised Associative Classification?

Validity of the findings

Which are the most advantages of your proposed method Self-Training Associative Classification using Ant Colony Optimization (ST-AC-ACO) in comparison with other existing techniques of fault classification? Could you please explain which are the differences between Semi-Supervised Learning and Supervised Learning algorithms and support an evidence of using Semi-Supervised Learning rather than Supervised Learning. Alternatively, could you please provide either some Tables or update simulation results to make an explanation the advantages of your proposed methodology ST-AC-ACO of fault classification and update some statistical criterion based on your research topic.

Additional comments

Recently, a new survey paper of wind turbine systems concentrates on fault diagnosis, prognosis and resilient control, which includes model-based, signal-based, and knowledge-based (data-driven) techniques to demonstrate the characteristics of fault detection, classification, and isolation in various faulty scenarios. In addition, some proposed methodologies on Semi-Supervised Learning and Ant Colony Optimisation have been addressed in the following journal papers.

If possible, could you please update any References for having in comparison with other algorithms in the specific field of your research topic for fault classification, as well as making any comments? These References are shown in the following below.

Reference 1: X. Wang, et. al., “EnAET: A Self-Trained Framework for Semi-Supervised and Supervised Learning With Ensemble Transformations,” in IEEE Transactions on Image Processing, vol. 30, pp. 1639–1647, 2021.

Reference 2: S. Mittal, et. al., “Semi-Supervised Semantic Segmentation With High- and Low-Level Consistency,” in IEEE Transactions on Pattern Analysis and Machine Intelligence, vol. 43, no. 4, pp. 1369–1379, April 2021.

Reference 3: “An Overview on Fault Diagnosis, Prognosis and Resilient Control for Wind Turbine Systems,” Processes, vol. 9, no. 2, p. 300, Feb. 2021.

Reference 4: W. -N. Chen, et. al., “Ant Colony Optimization for the Control of Pollutant Spreading on Social Networks,” in IEEE Transactions on Cybernetics, vol. 50, no. 9, pp. 4053–4065, Sep. 2020.

Reference 5: C. Juang, et. al., “Multiobjective Rule-Based Cooperative Continuous Ant Colony Optimized Fuzzy Systems With a Robot Control Application,” in IEEE Transactions on Cybernetics, vol. 50, no. 2, pp. 650–663, Feb. 2020.

Reference 6: “Actuator and Sensor Fault Classification for Wind Turbine Systems Based on Fast Fourier Transform and Uncorrelated Multi-Linear Principal Component Analysis Techniques,” Processes, vol. 8, no. 9, p. 1066, Sep. 2020.

---

## Round 0.2 · Minor Revisions

A reviewer still has concerns. As a result, a further revision is suggested.

Reviewer 2 has requested that you cite specific references, including one on which I am an author. Please be assured that I DO NOT expect you to include these citations and this will have no influence on my decision.

Reviewer 1 ·

Basic reporting

Paper looks good

Experimental design

I feel that this paper has the ability to deal with the real time on-field experimentation and the provided results are completely convincing.

Validity of the findings

The proposed scheme describes the author have a deep theoretical knowledge and very good orientation in the problem solving approach to delivered in the paper.

Additional comments

In the final version of revision, authors carefully addressed technical comments in the main text. I recommend acceptance on this round.

Reviewer 2 ·

Basic reporting

no comment

Experimental design

no comment

Validity of the findings

no comment

Additional comments

Which are the most advantages of your proposed method Semi-Supervised Associative Classification using Ant Colony Optimization Algorithm (SS-AC-ACOA) in comparison with other existing techniques of fault classification? Whether or not the following topic (https://doi.org/10.3390/pr9020300) can be discussed? Could you please explain which are the differences among Unsupervised Machine Learning (USML), Semi-Supervised Machine Learning (SSML), and Supervised Learning (SML) algorithms and support an evidence of using SSML rather than USML or/and SML. Alternatively, could you please provide either some TABLES or update Simulation or/and Experimental Results to make an explanation the advantages of your proposed methodology SS-AC-ACOA of fault classification and update some statistical criterion based on the research topic of this paper?

Additionally, the flowchart of proposed method SS-AC-ACOA mentioned in this paper should also be updated and included the dataset collections, pre-processing of dataset, post-processing of dataset, feature extractions, feature selections, and the performances of fault classification. Whether or not the following topic (https://doi.org/10.3390/pr8091066) can be discussed?

Specifically, it is worthy to point out, different topologies of experimental samples and the number of significant feature selections might have either positive or negative impacts for the performances of fault classification for more complicated industrial systems to some extent. Based on this reason, could you please provide an evidence to make explanations the availability, feasibility, and capability of the proposed methodology based on SS-AC-ACOA in the field of Semi-Supervised Associative Classification?

---

## Round 0.3 · accepted · Accept

The paper is ready to be accepted.